# Using synchronized brain rhythms to bias memory-guided decisions

**John J Stout[1], Allison E George[2], Suhyeong Kim[1], Henry L Hallock[3], Amy L Griffin[1]\***

[1]Department of Psychological and Brain Sciences, University of Delaware, Newark, United States; [2]Stony Brook University, Stony Brook, United States; [3]Neuroscience Program, Lafayette College, Easton, United States

**Abstract** Functional interactions between the prefrontal cortex and hippocampus, as revealed by strong oscillatory synchronization in the theta (6–11 Hz) frequency range, correlate with memory-guided decision-making. However, the degree to which this form of long-range synchronization influences memory-guided choice remains unclear. We developed a brain-machine interface that initiated task trials based on the magnitude of prefrontal-hippocampal theta synchronization, then measured choice outcomes. Trials initiated based on strong prefrontal-hippocampal theta synchrony were more likely to be correct compared to control trials on both working memory-dependent and -independent tasks. Prefrontal-thalamic neural interactions increased with prefrontal-hippocampal synchrony and optogenetic activation of the ventral midline thalamus primarily entrained prefrontal theta rhythms, but dynamically modulated synchrony. Together, our results show that prefrontal-hippocampal theta synchronization leads to a higher probability of a correct choice and strengthens prefrontal-thalamic dialogue. Our findings reveal new insights into the neural circuit dynamics underlying memory-guided choices and highlight a promising technique to potentiate cognitive processes or behavior via brain-machine interfacing.

## eLife assessment

This study enhances our understanding of the relationship between cortico-hippocampal interactions and behavioral performance. Using an inter-areal coherence metric to gate trial initiation in real time, the authors provide **solid** evidence that links high hippocampal-prefrontal theta coherence to correct performance on spatial working memory and cue-guided decision-making tasks. Although reviewers agreed that the results do not demonstrate causality between hippocampal-prefrontal synchrony and behavioral performance, the findings are viewed as **important** given their potential implications for brain-machine interface applications in humans.

## Introduction

Working memory, the ability to temporarily maintain and mentally manipulate information, is fundamental to cognition (*Baddeley, 1986*). This ability is known to require communication across distributed brain regions and is conserved over mammalia (*Goldman-Rakic, 1991*; *Sarnthein et al., 1998*; *Lee and Kesner, 2003*; *Winter and Stich, 2005*; *Wang and Cai, 2006*; *Eichenbaum, 2008*; *Fell and Axmacher, 2011*; *Christophel et al., 2017*; *Eichenbaum, 2017*; *Churchwell and Kesner, 2011*; *Spellman et al., 2015*; *Hallock et al., 2016*; *Ito et al., 2015*; *Bolkan et al., 2017*; *Ito et al., 2018*; *Maisson et al., 2018*; *Lugtmeijer et al., 2021*). Long-range interactions are thought to be supported by the proper timing of action potentials (spikes), and brain rhythms are thought to act as a clocking mechanism to synchronize the timing of spike discharges (*Fries, 2005*; *Buzsáki, 2006*;

**\*For correspondence:** amygriff@udel.edu

**Competing interest:** The authors declare that no competing interests exist.

**Preprint posted** 24 August 2023

**Sent for Review** 25 August 2023

**Reviewed preprint posted** 02 November 2023

**Reviewed preprint revised** 08 March 2024

**Version of Record published** 22 July 2024

*Fell and Axmacher, 2011*; *Colgin, 2011*; *Fries, 2015*). Fluctuations in the local field potential (LFP) are coupled to the organization of hippocampal spiking activity in rats (*O'Keefe and Recce, 1993*), primates (*Jutras et al., 2009*), and humans (*Qasim et al., 2021*), although the exact frequency can vary over mammalia. The hypothesis that brain rhythms coordinate brain communication by synchronizing neuronal activity, known as 'communication through coherence', is just beginning to be experimentally tested (*Fries, 2005*; *Buzsáki, 2006*; *Fell and Axmacher, 2011*; *Fries, 2015*; *Reinhart and Nguyen, 2019*).

In rats, decades of research have shown that computations within, and communication between, the medial prefrontal cortex (mPFC) and hippocampus are required for spatial working memory (*Dudchenko et al., 2000*; *Lee and Kesner, 2003*; *Wang and Cai, 2006*; *Horst and Laubach, 2009*; *Churchwell and Kesner, 2011*; *Hallock et al., 2013a*). Recording studies specifically implicate theta synchrony within the mPFC-hippocampal network as a mechanism for mPFC-hippocampal communication. One metric of oscillatory synchrony, coherence, has been repeatedly correlated with memory-guided choices (*Jones and Wilson, 2005*; *Benchenane et al., 2010*; *Sigurdsson et al., 2010*; *O'Neill et al., 2013*; *Hallock et al., 2016*), but also with attention and task engagement (*Guise and Shapiro, 2017*; *Bygrave et al., 2019*). In a cornerstone experiment, *Jones and Wilson, 2005*, showed that 4–12 Hz mPFC-hippocampal coherence was stronger before rats made a correct choice when compared to a choice error or a forced navigation trial on a spatial memory task. Importantly, these results are derived from measurements of magnitude squared coherence, a measurement of signal correlation, with no requirement for exact numerical phase consistency. For example, two structures can exhibit strong magnitude squared coherence, despite two signals being approximately 180 degree offset in phase. This is an important distinction because there currently exist two versions of the communication through coherence hypothesis; first, that inter-areal communication varies with signal phase, irrespective of coherence, and second that inter-areal communication varies with coherence (*Vinck et al., 2023*). Likewise, the finding that mPFC-hippocampal theta coherence was stronger on correct choice outcomes is potentially conflated with the fact that rodent movement behaviors also change with task performance (*Redish, 2016*). Due to constraints on experimental design, it remains unclear as to whether strong theta coherence increased the likelihood of a correct choice, or whether a correct choice led to stronger theta coherence. Addressing this question is of critical importance for the potential use of oscillatory dynamics in therapeutic settings (*Reinhart and Nguyen, 2019*).

We hypothesized that if magnitude squared coherence represents a valid mechanism for prefrontal-hippocampal communication, that we could use times of strong mPFC-hippocampal theta coherence to gate access to the choice, and that these trials would be associated with better performance on memory-guided tasks. To circumvent a purely correlational experimental design, we developed programmatic algorithms to define and detect strong and weak oscillatory synchronization, then tied theta (6–11 Hz) coherence magnitude with task manipulation. This brain-machine interface monitored details about task trials, like delay duration and choice outcome, while dynamically adjusting future trials to serve as within-subject controls. Trials initiated during times of strong mPFC-hippocampal theta coherence were associated with correct choice outcomes on both spatial working memory-dependent and -independent tasks. In follow-up experiments, we found that mPFC theta rhythms and mPFC-thalamic interactions increased with mPFC-hippocampal theta synchrony. Consistent with these results, optogenetic activation of the ventral midline thalamus (VMT), a structure known to coordinate mPFC-hippocampal interactions (*Vertes, 2002*; *McKenna and Vertes, 2004*; *Gabbott et al., 2005*; *Vertes et al., 2006*; *Hoover and Vertes, 2007*; *Hoover and Vertes, 2012*; *Hallock et al., 2016*; *Dolleman-van der Weel et al., 2019*; *Griffin, 2021*), dynamically modulated mPFC and hippocampal theta oscillation power and coherence.

## Results
### Development of a closed-loop brain-machine interface for coherence-dependent task manipulation

Our first objective was to design and implement a brain-machine interface that would time the initiation of task trials to periods of strong or weak prefrontal-hippocampal theta synchronization (*Figure 1A*, *Figure 1—figure supplements 1–3*). To do this, we first trained rats to perform a delayed spatial alternation task in a T-maze until reaching 70% choice accuracy on 2 consecutive days. On this

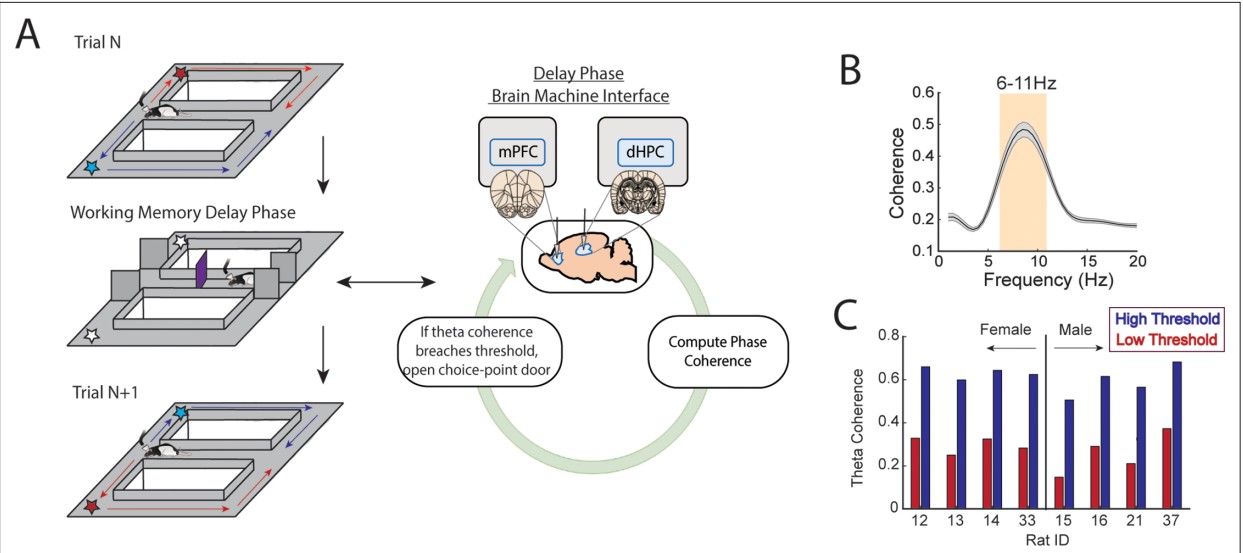

**Figure 1.** A brain-machine interface that harnesses endogenous medial prefrontal cortex (mPFC)-hippocampal theta coherence on a working memory task. (**A**) Schematic of brain-machine interfacing as rats performed a delayed alternation task on an automated T-maze. The delayed alternation task requires rats to alternate between left and right reward zones. Blue arrows and stars denote correct (rewarded) trajectories while red arrows and stars represent incorrect (unrewarded) trajectories. The rat was confined to the delay zone with three barriers. On a subset of trials, we computed mPFC-hippocampal theta coherence in real time during the delay and trials were initiated contingent upon theta coherence magnitude. (**B**) Frequency by coherence distribution calculated on data collected in real time. For brain-machine interfacing experiments, theta coherence was defined as the averaged coherence values between 6 and 11 Hz. Data are represented as the mean ± s.e.m. (**C**) Thresholds for high and low magnitude coherence were estimated based on distributions of theta coherence values that were unique to individual rats (see *Figure 1—figure supplement 2I and J* and Methods). *N*=8 rats (4 female, 4 male).

The online version of this article includes the following figure supplement(s) for figure 1:

**Figure supplement 1.** Two independent loops support brain-machine interfacing.

**Figure supplement 2.** Brain-machine interface parameterization.

**Figure supplement 3.** Detailed representation of brain-machine interfacing.

**Figure supplement 4.** The effect of delay duration on delayed alternation task performance.

spatial working memory task, rats are rewarded for alternating between left and right reward zones and sequestered at the base of the maze before each choice (*Figure 1A*). The ability of this task to tax working memory was validated by measuring the impact of delay duration on choice outcome. Consistent with the use of delayed-response tasks across species (*Dudchenko, 2004*; *Goldman-Rakic, 1991*; *Eichenbaum, 2008*), longer delay durations were associated with lower choice accuracy (*Figure 1—figure supplement 4A*).

Rats were implanted with stainless steel wires targeting the prelimbic and infralimbic subregions of the mPFC and CA1 of dorsal hippocampus (dHPC) (*Figures 1A and 2A*) to record LFPs. During training sessions, thousands of theta coherence values were calculated during the delay phases, and distributions of mean theta coherence estimates were created (*Figure 1—figure supplement 2J*). Using these distributions, we defined weak theta coherence as 1 std below the mean, and strong theta coherence as 1 std above the mean of all theta coherence values. Therefore, each rat had a unique numerical value defining states of strong and weak theta coherence (*Figure 1C*), which we could then use as thresholds to initiate trials on the automated maze. Trial initiation is defined by the lowering of the choice point door to allow access to the maze (*Figure 1A*; *Figure 1—figure supplement 3*).

To support brain-machine interfacing (see Methods section 'Brain-machine interface'), we designed two independent loops, one processing the neural data in real time and the other controlling the automatic maze (*Figure 1—figure supplement 1*; *Figure 1A*). This closed loop system allowed us to monitor prefrontal-hippocampal theta coherence in real time and on a subset of trials, initiate the start of the trial when coherence was strong or weak. While coherence was being monitored, rats were confined to an area at the base of the maze. Trials were initiated by opening a door, providing access to the maze (*Figure 1A*).

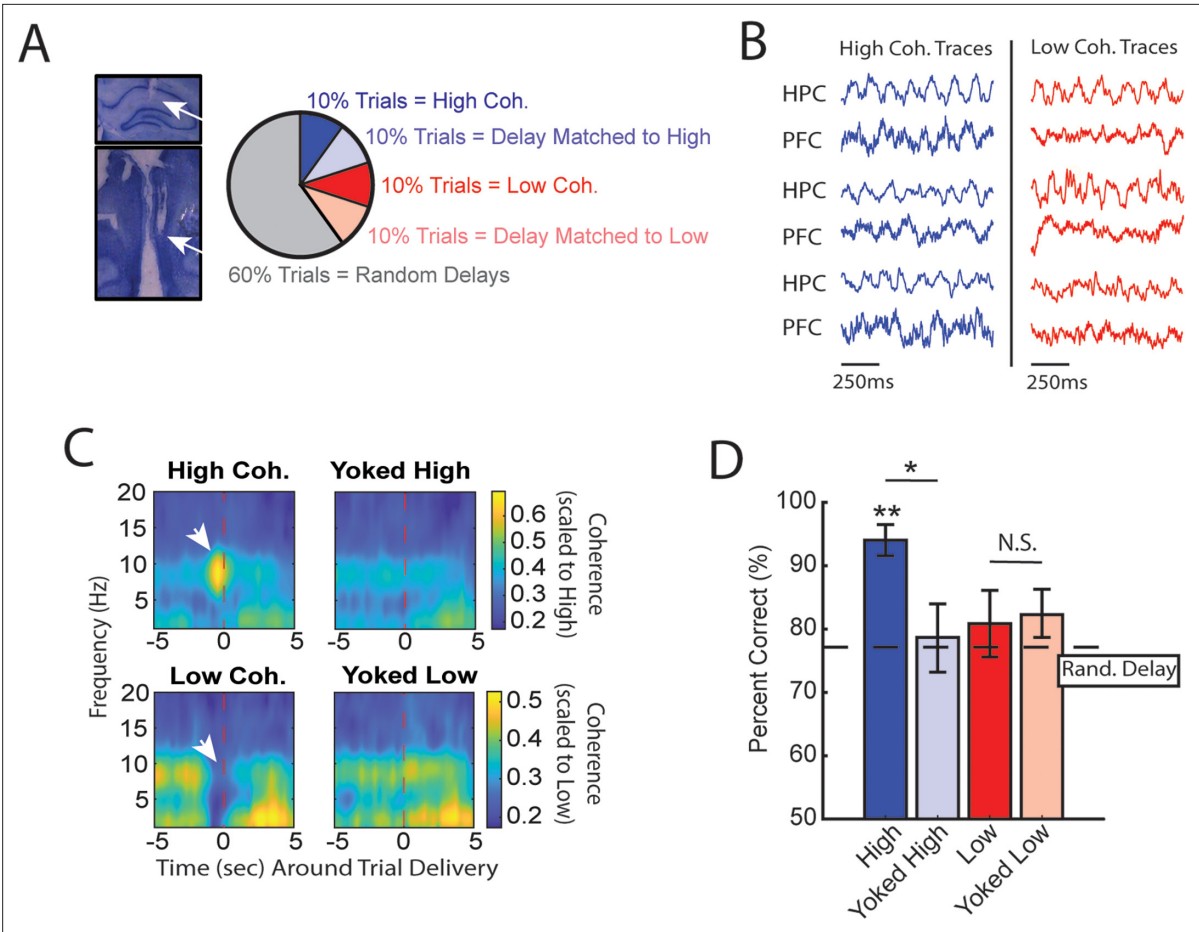

**Figure 2.** High medial prefrontal cortex (mPFC)-hippocampal theta coherence can be used to enhance performance of a working memory-dependent task. (**A**) *Left panel:* Histology from a representative rat showing electrode tracks in the dorsal hippocampus (*top*) and mPFC (*bottom*). *Right panel:* Distribution of trial types within a session. Within 10-trial blocks, 20% of trials were initiated based on high or low mPFC-hippocampal theta coherence, 20% of trials were yoked to the high/low coherence trials, and 60% were triggered following a random delay (5–30 s). Yoked trials were identical in delay duration as high/low coherence trials, but triggered independent of coherence magnitude to control for the negative correlation between delay length and task performance (*Figure 1—figure supplement 4*). (**B**) Example local field potential (LFP) traces recorded during high and low coherence trials from three representative rats. The mPFC and hippocampal signals were used to compute theta coherence in real time. (**C**) Rat-averaged coherograms representing time around trial initiation (*x*-axis), coherence frequency (*y*-axis), and coherence magnitude, with warmer colors indicating higher coherence values. White arrows denote strong (top panel) and weak (bottom panel) theta coherence, as expected on trials triggered during high and low coherence states. Notice that on yoked trials, coherence was rather consistent before and after trial initiation, as expected for trials triggered independent of coherence magnitude. (**D**) Relative to yoked trials, presenting choices to rats when mPFC-hippocampal theta coherence was high led to improved task performance (t(7) = 2.85, $p_{p.c.}$=0.0248). Trials contingent upon low magnitude theta coherence did not impact task performance compared to delay matched controls (t(7) = –0.26, $p_{p.c.}$=0.80; paired t-test). Follow-up statistical testing revealed that choice accuracy on high coherence trials was significantly greater than choice accuracy on random delays, consistent with our planned comparisons between high and yoked trials (t(7) = 6.12; $p_{(x4)}$=0.002). See *Supplementary file 1a* for statistics. *p<0.05, **p<0.01. Stars (**) above bar graph denotes significance as measured from comparisons relative to random delay choice outcomes (black) and relative to 70% criterion (gray). Subscript 'p.c.' indicates planned comparisons. Subscript '(x4)' indicates unplanned comparisons with Bonferroni corrected p-values for the number of unplanned tests performed. *N*=8 rats (4 male, 4 female).

The online version of this article includes the following figure supplement(s) for figure 2:

**Figure supplement 1.** Analysis of random delay trial onset with coincident strong medial prefrontal cortex (mPFC)-hippocampal theta coherence.

**Figure supplement 2.** Behavioral analyses from the brain-machine interfacing experiment in *Figure 2*.

## Strong prefrontal-hippocampal theta coherence leads to correct choices on a spatial working memory task

Based on multiple reports, mPFC-hippocampal theta coherence is positively correlated with memory-guided decision-making (*Jones and Wilson, 2005*; *Benchenane et al., 2010*; *Hallock et al., 2016*), but whether theta coherence can be harnessed to bias choice accuracy remains unexplored. To test

this idea, we implemented the algorithms described above with an automatic maze to control trial onset via lowering the door for access to the choice (*Figure 1A*; *Figure 1—figure supplements 1 and 3*). During experimentation, our brain-machine interface was activated as rats occupied the delay zone and rats were presented with various trial types within a given session as follows. A small proportion of trials were initiated when mPFC-hippocampal theta coherence was above the strong theta coherence threshold (~10% of trials) or below the weak theta coherence threshold (~10% of trials) (*Figure 2A and B*). Since increasing delay durations led to worse task performance (*Figure 1—figure supplement 4*), rats also experienced trials that were yoked to high and low coherence trials via identical delay durations. For example, if trial *N* was a high coherence trial, our algorithm logged the duration spent in the delay zone to be presented back to the rat within a 10-trial block. Thus, initiation of yoked trials was independent of the strength of theta coherence (*Figure 2C*) and by comparing choice accuracy on strong/weak coherence trials to that on yoked trials, we were able to rule out the possible confounding variable of working memory load on choice accuracy.

We predicted that, relative to yoked control trials, trials presented during states of strong mPFC-hippocampal theta coherence would be more likely to be correct and trials presented during states of weak mPFC-hippocampal theta coherence would be more likely to be incorrect. Consistent with our first prediction, presenting trials during elevated states of mPFC-hippocampal theta coherence improved choice accuracy (*Figure 2D*). However, choice accuracy on trials presented during states of low mPFC-hippocampal theta coherence did not differ from choice accuracy on yoked control trials, indicating that naturally occurring weak theta synchronization does not impair choice outcomes.

Most task trials (~80%) were initiated after a random delay, irrespective of the magnitude of mPFC-hippocampal theta coherence. We next analyzed whether random delay trials that were coincident with strong mPFC-hippocampal theta coherence also led to correct choice outcomes. First, compared to brain-machine interfacing trials, random delay trials coincident with strong mPFC-hippocampal theta coherence were found to be significantly longer in duration (*Figure 2—figure supplement 1A*; BMI trials: mean = 11.55 s, std = 1.51 s; random trials with strong theta coherence: mean = 15.5 s, std = 2.2 s), an important finding because task performance is impacted by time spent in the delay (*Figure 1—figure supplement 4*). Unlike brain-machine interfacing trials, which had yoked conditions built into 10-trial blocks to account for changing behavior over time, random delay trials that were triggered during strong mPFC-hippocampal theta coherence states were not programmed to have a control. As such, we approximated a yoked condition by identifying random delay trials with identical delay durations as random delay trials with high theta coherence (*Figure 2—figure supplement 1B and C*). These trials were distributed throughout the session and were unequal in contribution (i.e. there may exist multiple 7 s trials to match a 7 s random trial with coincident strong theta coherence). Although there was no significant difference between random delay trials coincident with strong theta coherence compared to trials with identical delay durations (p=0.059; *Figure 2—figure supplement 1B*), 6/8 animals showed better task performance when mPFC-hippocampal theta coherence was strong (*Figure 2—figure supplement 1D*). Given that this comparison is fundamentally different from the brain-machine interfacing experiment due to imbalanced design between estimated yoked trials and random trials with high coherence, and did not account for trials with potential salient/distracting events in the environment, we consider these results consistent with our brain-machine interfacing findings.

We then examined various measurements of overt behavior to test if behaviors differed between coherence-triggered trials and yoked trials. First, we examined the amount of time spent until rats made a choice, defined as the amount of time from the point at which a trial is initiated until rats passed the infrared beam that triggers the reward dispenser (*Figure 2—figure supplement 2*). While we found no difference in time to choice between high coherence trials and yoked trials, there was a trending difference between low and yoked trials (*Figure 2—figure supplement 2A*). Using an analysis to test head-movement complexity (IdPhi; *Papale et al., 2012*; *Redish, 2016*), we found no differences between high coherence trials and yoked trials but did observe less head-movement complexity on low coherence trials relative to yoked trials (*Figure 2—figure supplement 2B*). Next, we analyzed total distance traveled in the epoch used to trigger trials during high and low coherence states (last 1.25 s before trial initiation). Since the amount of time was always consistent (1.25 s), this approach is a proxy for speed, an indirect correlate of theta frequency (*Kropff et al., 2021*). We found no differences in movement behavior between coherence trials and yoked trials (*Figure 2—figure*

*supplement 2C*). Finally, we found that rats spent similar amounts of time in the delay zone during high and low coherence trials (*Figure 2—figure supplement 2D*). These analyses show that high coherence trials could be used to promote correct choices in the absence of overt differences in behavior between trial types, indicating that mPFC-hippocampal theta coherence preceding the choice potentially influences choice outcome.

## Trials initiated by strong prefrontal-hippocampal theta coherence are characterized by prominent prefrontal theta rhythms and heightened pre-choice prefrontal-hippocampal synchrony

Next, we performed offline data analysis to understand the neural dynamics occurring during the high coherence states that improved spatial working memory task performance. First, we noticed that theta rhythms were better characterized by changes within the 6–9 Hz range (*Figure 3A*) and as such, offline analyses focused on this narrow band. Relative to low coherence states, mPFC theta rhythms were stronger during high coherence states (*Figure 3A and B*; see *Figure 2B* for example LFP traces). Hippocampal theta rhythms only exhibited a modest elevation in theta power relative to low coherence states. With respect to theta frequency, mPFC theta rhythms were shifted toward higher frequencies during high coherence states (mean theta frequency = 5.8 Hz) relative to low coherence states (mean theta frequency = 5 Hz) (*Figure 3C*). While there was no significant difference in hippocampal theta frequency, 6/8 rats showed higher theta frequency during high mPFC-hippocampal theta coherence states (mean theta frequency during high coherence states = 7 Hz; mean theta frequency during low coherence states = 6.5 Hz). We then analyzed whether these signals exhibited evidence of directionality, the ability for one signal to predict another signal as measured by Granger causality analysis (*Cohen, 2014*). Relative to low coherence states, high coherence states were characterized by stronger hippocampal-to-mPFC theta directionality (*Figure 3D*). Thus, the high theta coherence states used to trigger spatial working memory trials were characterized by strong mPFC theta rhythms and hippocampal-to-mPFC theta directionality.

Even though the delay zone was physically close to the choice point (~30 cm), we wondered whether strong mPFC-hippocampal theta coherence trials impacted synchronization during the goal-arm choice. Therefore, we defined choice point entry as the infrared beam break immediately preceding the choice (*Figure 1—figure supplement 1*). On average, rats took 1.6 and 2.1 s to reach this infrared beam from trial initiation on low and high coherence trials, respectively. No significant difference in time to choice was observed between high and low coherence trials (*Figure 3E*). Thus, we extracted LFPs from –2 to +0.5 s surrounding choice entry (*Figure 3E*), and calculated coherence over time and frequency (*Figure 3F*). A normalized difference score was calculated from the resultant coherograms (high-low/high+low), revealing a clear difference in theta coherence magnitude between high and low coherence trials as rats approached the choice zone (*Figure 3G*). As expected, high coherence trials showed significantly stronger synchronization at –2 s, an approximate for trial initiation (*Figure 3H*). Interestingly, after the 2 s time point, theta coherence between high and low coherence trials became more similar, but once again differed at ~0.4–0.5 s pre-choice and post-choice entry (*Figure 3H*). This latter result shows that strong mPFC-hippocampal theta coherence during the delay was maintained throughout choice.

We observed mPFC-hippocampal theta coherence to fluctuate rhythmically (*Figure 1—figure supplement 2H*; *Figure 3—figure supplement 1B*), and therefore wondered how predictive past values of mPFC-hippocampal theta coherence were of future values. Using previously collected data (*Hallock et al., 2016*), we extracted mPFC-hippocampal theta coherence epochs across the duration of a 30 s delay on the delayed alternation (DA) task from three rats (*N*=22 sessions; *Figure 3—figure supplements 1A and 2A*). We performed an autocorrelation analysis on theta coherence values on a trial-by-trial basis, then compared the results to a temporally shuffled theta coherence distribution. Since we performed a moving window approach (1.25 s in 250 ms steps), comparisons between real and temporally shuffled coherence estimates were only included after five lags (lag #4 relative to 0; *Figure 3—figure supplement 1C*). While theta coherence values were predictive of future theta coherence values, this effect slowly decayed over time, indicating that despite some observations of periodicity, the fluctuations were largely non-periodical (*Figure 3—figure supplement 1C*).

In our brain-machine interfacing experiments, trials were initiated when mPFC-hippocampal theta coherence was strong or weak. States of strong mPFC-hippocampal theta coherence

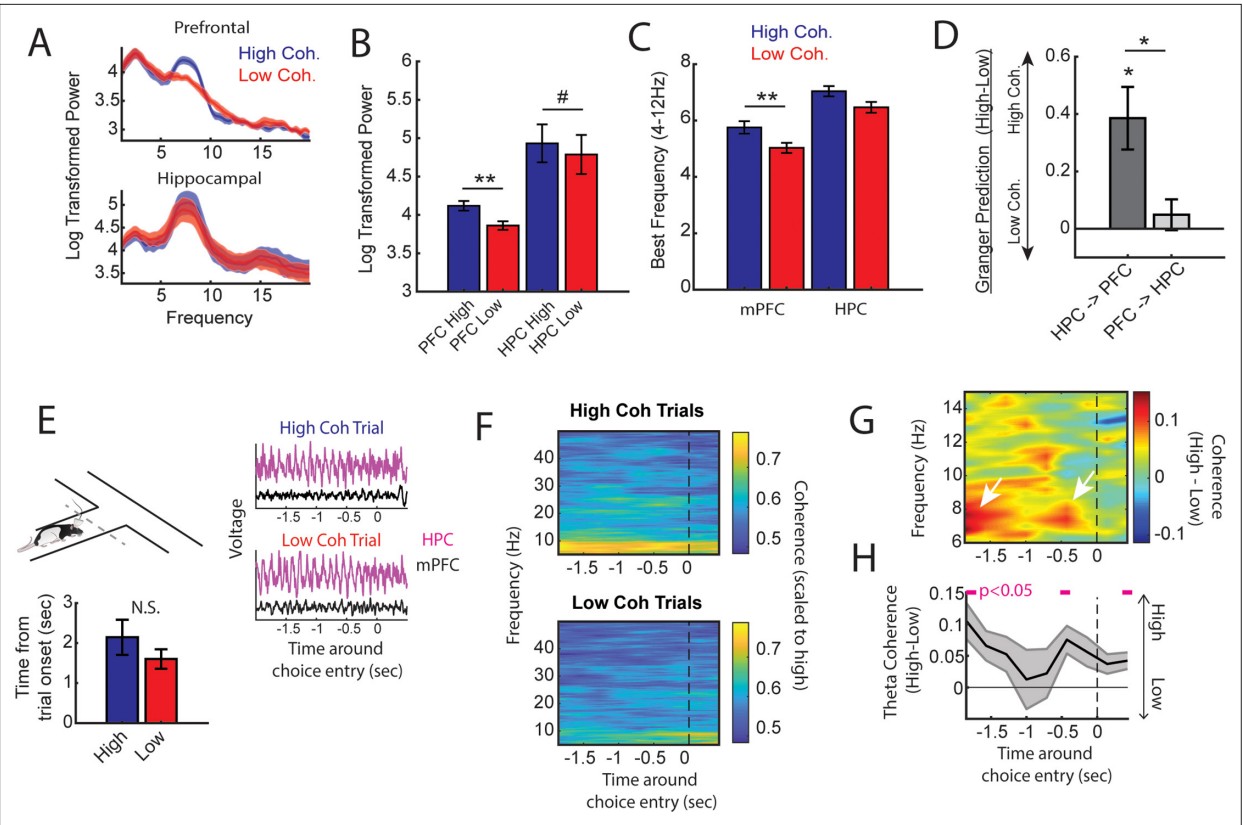

**Figure 3.** High medial prefrontal cortex (mPFC)-hippocampal theta coherence trials are gated by prefrontal theta rhythms and lead to heightened pre-choice synchrony. (**A**) Prefrontal and hippocampal power spectra during the high and low coherence epochs used for brain-machine interfacing (*Figures 1 and 2*). (**B**) Prefrontal theta power (6–9 Hz) was significantly greater during high coherence epochs relative to low coherence epochs (t(7) = 5.3, ci = 0.14–0.37, $p_{adj(x2)}$=0.002). Hippocampal theta power was stronger on high coherence compared to low coherence trials (t(7) = 2.47, ci = 0.006–0.28, $p_{adj(x2)}$=0.08, $p_{not-adj}$=0.0427). (**C**) The frequency of prefrontal theta oscillations was significantly higher during high coherence states relative to low coherence states (PFC: t(7) = 3.08, $p_{adj(x2)}$=0.036, ci = 0.16–1.3; hippocampus: t(7) = 1.8, ci = –0.17 to 1.3, p=0.11). Note that 6/8 rats showed higher theta frequency in the hippocampus on high theta coherence states relative to low theta coherence states. Theta frequency was measured by identifying the frequency corresponding to maximum theta power. (**D**) Hippocampal-to-prefrontal theta directionality was significantly stronger during high theta coherence states relative to low theta coherence states (t(7) = 3.53, ci = [0.12–0.64], $p_{adj(x3)}$=0.029) and was significantly stronger than Granger prediction in the prefrontal-to-hippocampal direction (t(7) = 3.33, ci = 0.097–0.57, $p_{adj(x3)}$=0.038). No significant effect was observed in the prefrontal-hippocampal direction (t(7) = 0.909, p=0.39). (**E**) Local field potential (LFP) signals (jittered for visualization) were extracted from 2 s before choice point entry (as defined by infrared beam breaks) and 0.5 s afterward. Bar graphs show that the average time to choice entry for high coherence and low coherence trials was between 1.6 and 2.1 s and did not significantly differ between trial types (t(7) = 2.0, p=0.08). (**F**) Averaged coherograms (N=8 rats) showing coherence as a function of frequency and time surrounding choice point entry. (**G**) Difference of the coherograms shown in **F**. White arrows point to initial 6–9 Hz synchronization at –2 s which approximates trial onset (see bar graph in **E**), and a second time point of heightened theta synchrony before choice entry. (**H**) Normalized difference scores representing theta coherence as a function of time. Theta coherence at choice entry was significantly stronger on trials triggered by high coherence relative to trials triggered during low coherence (see *Supplementary file 1b* for raw and corrected p-values). Data are represented as the mean ± s.e.m. across eight rats. *p<0.05, **p<0.01 paired t-tests with Bonferroni p-value corrections when p<0.05. Difference scores were tested against a null of 0. Magenta lines denote p<0.05 after Benjamini-Hochberg corrections.

The online version of this article includes the following figure supplement(s) for figure 3:

**Figure supplement 1.** Medial prefrontal cortex (mPFC)-hippocampal theta coherence across a fixed delay.

**Figure supplement 2.** Details regarding mPFC-VMT-HPC recordings.

increased the probability of a correct choice, while increasing synchronization during task performance. However, when we examined the frequency of strong mPFC-hippocampal theta coherence events when the delay phase was fixed and predictable, strong mPFC-hippocampal theta coherence events did not predict trial initiation (*Figure 3—figure supplement 1D*). When considered with the results above, mPFC-hippocampal theta coherence events predict choice outcome, rather than trial onset.

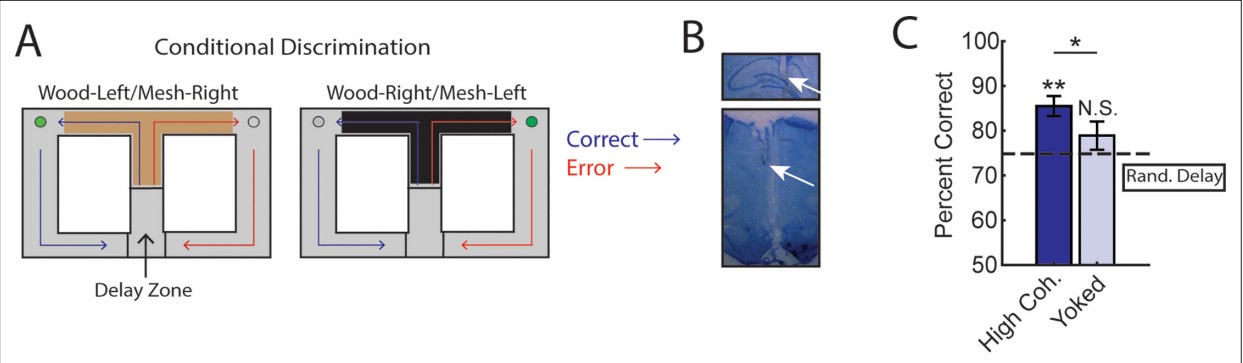

**Figure 4.** Trials initiated by strong medial prefrontal cortex (mPFC)-hippocampal theta coherence enhance task performance on a two-choice conditional discrimination task. (**A**) Schematic of the conditional discrimination task. Wooden or mesh floor inserts were used to guide choice behavior. Rats were randomly assigned to insert-reward contingencies. Like the brain-machine interfacing experiment on the delayed alternation task, trials were initiated when rats were sequestered in the delay zone. (**B**) Example histology from a representative rat showing electrode placements in the hippocampus and mPFC. (**C**) Trials initiated during high mPFC-hippocampal theta coherence states led to better task performance when compared to yoked control trials (t(15) = 2.23, ci = 0.29–12.87, $p_{(p.c.)}$=0.04) or when compared to trials triggered following a random delay (t(15) = 3.8, ci = 4.7–16.6, $p_{(x2)}$=0.002). There was no difference in choice outcome following yoked and random delay trials (t(15) = 1.0, ci = –4.5 to 12.7, $p_{(x2)}$=0.33). *p<0.05. **p<0.01. Subscript on p-values shows if comparisons were planned ('p.c.') or corrected for multiple comparisons ('x2'). Data are represented as the mean ± s.e.m. $N$=16 sessions over three rats.

## Prefrontal-hippocampal theta coherence states lead to correct choices on a conditional discrimination task

Our findings from *Figure 2* show that mPFC-hippocampal theta coherence leads to correct spatial working memory-guided choices. We next wondered if this effect was specific to spatial working memory and specifically tested whether strong mPFC-hippocampal theta coherence events were optimal for choices on a task where rats must attend to external stimuli to guide decision-making. Rats ($N$=3; 1 male, 2 female) were implanted with wires targeting the mPFC and hippocampus (*Figure 4B*) and were trained to perform a conditional discrimination task where a floor insert dictated choice outcome (e.g. a wooden floor insert signals a left choice, while a mesh insert signals a right choice; *Figure 4A*). This task is similar in difficulty to the DA task, but requires the dorsal striatum, rather than the hippocampus to perform (*Hallock et al., 2013a*). Likewise, past research showed that inactivation of the mPFC or the VMT did not disrupt conditional discrimination task performance in well-trained rats (*Hallock et al., 2013b*; *Shaw et al., 2013*), indicating that the mPFC-hippocampal network is not required for conditional discrimination task performance. Therefore, we predicted that strong mPFC-hippocampal theta coherence would not improve choice outcomes on this conditional discrimination task.

We collected 35 sessions, of which 16 sessions (7 sessions from 21 to 48 [male]; 4 sessions from 21 to 49 [female]; and 5 sessions from 21 to 55 [female]) met criterion for performance of >70%, alternation of <70%, and a contribution of at least three trials. Unexpectedly, we found that initiation of trials during strong mPFC-hippocampal theta coherence enhanced choice accuracy on the conditional discrimination task (*Figure 4C*). This finding was surprising given that mPFC-hippocampal theta coherence did not previously correlate with choice outcomes on the conditional discrimination task (*Hallock et al., 2016*), but consistent with increased mPFC-hippocampal theta coherence on a different cue-guided paradigm (*Benchenane et al., 2010*). Most importantly, these results show that strong mPFC-hippocampal theta coherence is optimal for decision-making behavior regardless of whether working memory and mPFC/hippocampal function is necessary to perform a task.

## Prefrontal-thalamo-hippocampal network dynamics vary with prefrontal-hippocampal synchronization

So far, we have shown that initiating trials when mPFC-hippocampal theta synchrony is strong leads to correct memory-guided choices. What are the mechanisms supporting strong mPFC-hippocampal theta synchrony leading to improved choice accuracy? Past research showed that mPFC-hippocampal theta synchrony during choice was supported by the VMT (*Hallock et al., 2016*). The VMT is

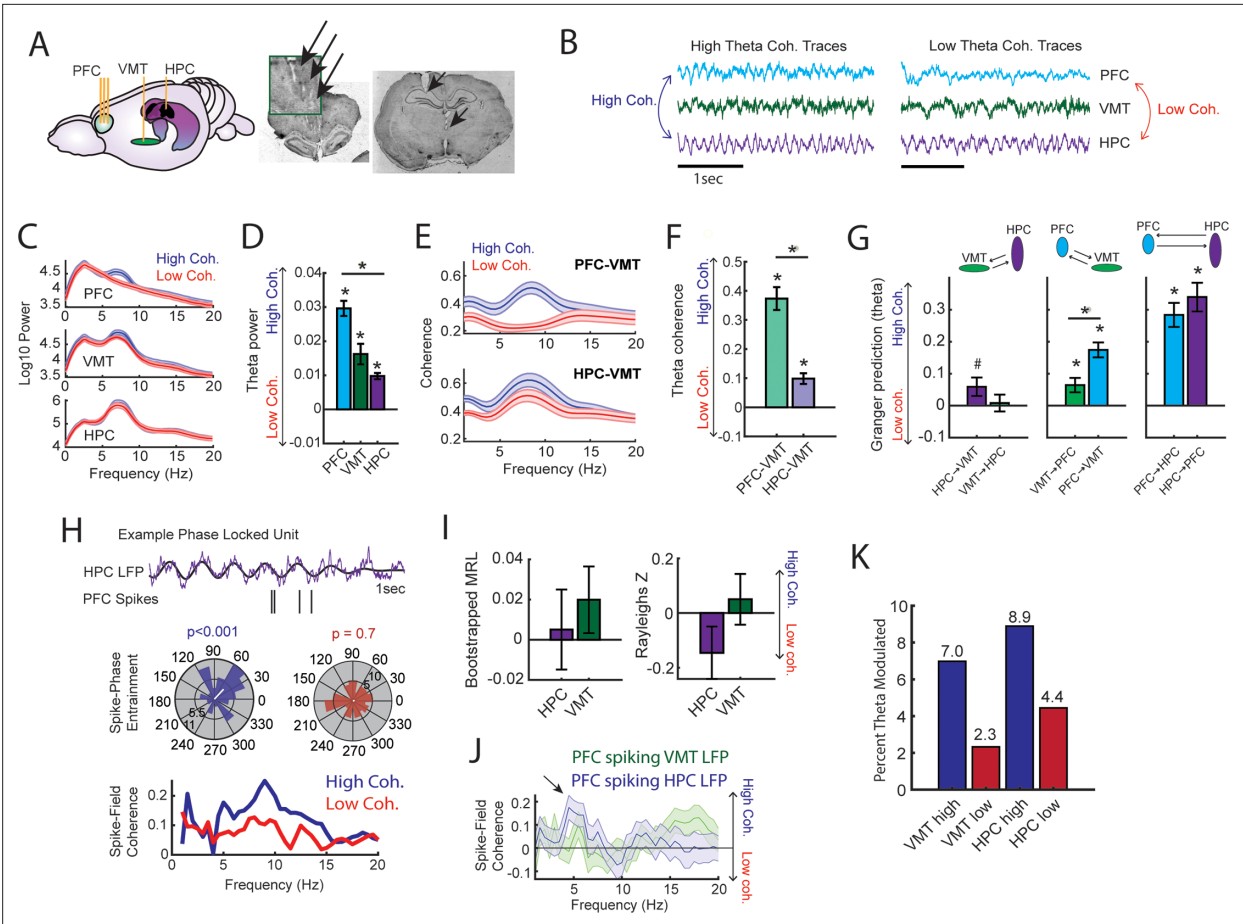

**Figure 5.** Prefrontal-hippocampal theta synchronization modulates prefrontal-thalamic interactions. (**A**) Local field potentials (LFPs) were recorded from the medial prefrontal cortex (mPFC), ventral midline thalamus (VMT), and hippocampal of three rats (*N*=22 sessions). *Right panel* shows triple site recordings taken from a representative rat. Green box shows example tetrode tracks from the mPFC. (**B**) High and low mPFC-hippocampal theta coherence epochs were identified, and LFP from the VMT was extracted. The data shown are collapsed across high or low coherence epochs. (**C**) Frequency by coherence plots from the mPFC (top panel), VMT (middle panel), and hippocampus (bottom panel). Compare these data to *Figure 3*. (**D**) Normalized difference scores comparing theta (6–9 Hz) power between high and low coherence epochs. There was a main effect of brain region on the coherence difference score (*F*(2,65) = 20.8; p<0.001; one-way ANOVA) with each brain area showing higher theta power during high coherence states relative to low coherence states (PFC: p<0.001; VMT; p<0.001; HPC: p<0.001; see *Supplementary file 1c*). (**E**) Theta coherence for mPFC-VMT and VMT-HPC was estimated during high and low mPFC-hippocampal theta coherence states. (**F**) mPFC-VMT and VMT-HPC theta coherence was stronger during high when compared to low mPFC-hippocampal theta coherence states. mPFC-VMT theta coherence changed more drastically with mPFC-hippocampal theta coherence magnitude (mPFC-VMT: p<0.001; VMT-HPC: p<0.001; mPFC-VMT vs VMT-HPC: p<0.001; see *Supplementary file 1d*). (**G**) Multivariate Granger prediction analysis. *Left panel* shows VMT-HPC theta directionality. *Middle panel* shows mPFC-VMT theta directionality. *Right panel* shows mPFC-hippocampal theta directionality. Granger prediction in the mPFC-to-VMT direction was more sensitive to mPFC-hippocampal theta coherence magnitude when compared to Granger prediction in the VMT-to-mPFC direction (statistics in *Supplementary file 1e*). (**H**) *Top panel* shows hippocampal LFP (1 s) and example spikes from an mPFC neuron with significant spike-theta entrainment. *Middle panel* shows polar plots of the unit in the top panel. Histogram represents the distribution of spike-phase values with the mean result length vector shown as a white bar in the center. *Bottom panel* shows spike-field coherence for the same neuron. (**I**) Difference score (high-low/high+low) of bootstrapped MRL and Rayleighs Z-statistic for each neuron as a function of hippocampal or VMT theta. No significant differences were found between high and low mPFC-hippocampal theta coherence states. (**J**) Spike-field coherence, represented as a difference score. No effects survived p-value correction. Arrow points to a numerical increase to spike-field coherence at hippocampal 4–6 Hz. (**K**) Percentage of significantly modulated mPFC units to VMT theta and hippocampal theta as a function of strong (blue) or weak (red) mPFC-hippocampal theta coherence states. *p<0.05. Data are represented as the mean ± s.e.m.

anatomically connected with the mPFC and hippocampus (*Sesack et al., 1989*; *Vertes, 2002*; *McKenna and Vertes, 2004*; *Vertes et al., 2006*; *Hoover and Vertes, 2007*; *Hoover and Vertes, 2012*), providing a source of glutamatergic excitation to both structures (*Dolleman-van der Weel et al., 2019*). Therefore, we wondered how mPFC-VMT and VMT-hippocampal interactions varied with mPFC-hippocampal theta synchronization.

To probe this question, we examined datasets with simultaneous mPFC, VMT, and dHPC recordings from three rats performing a spatial working memory task (*N*=22/28 sessions; *Figure 5A*; *Figure 3—figure supplement 2B*; *Stout and Griffin, 2020*). We extracted neural data as rats occupied the delay zone, then defined and detected epochs of strong and weak mPFC-hippocampal theta coherence offline (*Figure 5B*; *Figure 3—figure supplement 1A and B*; *Figure 3—figure supplement 2B*). Corroborating the findings from our brain-machine interfacing experiment (*Figures 2 and 3*), high theta coherence states were characterized by strong 6–9 Hz theta rhythms in the mPFC (*Figure 5C and D*). Intriguingly, the magnitude change of theta power between high and low coherence states was strongest in the mPFC, followed by the VMT, then the hippocampus (*Figure 5D*). Relative to low coherence epochs, the mPFC was differentially and simultaneously synchronized to the VMT and hippocampus during high coherence states (*Figure 5E*). Moreover, high coherence states were characterized by a stronger change in neural synchronization between the mPFC and VMT, relative to the VMT and hippocampus (*Figure 5F*). This latter result suggested that mPFC-VMT interactions may be particularly sensitive to mPFC-hippocampal synchronization. In support of this conclusion, multivariate Granger prediction revealed that mPFC-VMT directionality was elevated during strong relative to weak mPFC-hippocampal theta coherence states (*Figure 5G*; *middle panel*). mPFC-hippocampal directionality was also modulated by mPFC-hippocampal theta coherence magnitude. However, directionality between the VMT and hippocampus was minimally impacted by the magnitude of mPFC-hippocampal theta coherence (*Figure 5G*).

Lastly, we examined whether mPFC spike-LFP synchrony was impacted by mPFC-hippocampal theta coherence. Spike-phase entrainment was used to quantify the non-uniformity of spike-phase distributions at theta to measure theta phase locking, and spike-field coherence was used to understand the correlation between spikes and LFP across frequencies (*Figure 5H*). Out of 126 mPFC neurons, 46 neurons met criterion for inclusion (see Methods). When comparing strong to weak mPFC-hippocampal theta coherence states, there were no significant differences to theta phase entrainment (*Figure 5I*) nor to spike-field coherence (*Figure 5J*) of mPFC spikes to VMT and hippocampal theta.

We then wondered if strong mPFC-hippocampal theta coherence states modulated the spike timing of a select group of mPFC neurons. During strong mPFC-hippocampal theta coherence states, 8.9% and 7% of mPFC neurons were modulated by hippocampal theta and VMT theta, respectively. This contrasted with weak mPFC-hippocampal theta coherence states, where 4.4% and 2.3% of mPFC neurons were significantly modulated by hippocampal and VMT theta, respectively (*Figure 5K*). These findings indicate that the magnitude of mPFC-hippocampal theta synchronization was unrelated to global changes to mPFC spike entrainment to VMT and hippocampal theta rhythms. Instead, relative to low coherence states, states of high mPFC-hippocampal theta coherence were associated with strong mPFC spike-phase locking to VMT and hippocampal theta rhythms in a small group of mPFC neurons.

## Optogenetic activation of the VMT dynamically regulates prefrontal-hippocampal theta rhythms

Next, we examined whether artificial theta frequency stimulation of the VMT was sufficient to produce synchronized theta rhythms between the mPFC and hippocampus. To investigate this question, we injected the VMT with AAV5-hSyn-ChR2-eYPF to create and embed channelrhodopsin2 at the membrane of VMT neurons, a light-gated cation channel that promotes excitation of neurons with blue light stimulation (450 nm). This injection was combined with simultaneous recordings from the mPFC and hippocampus, as well as a fiber placed in the VMT (*Figure 6A*). After 4–6 weeks of recovery to allow for viral expression, we pulsed a blue laser targeting the VMT while recording from the mPFC (*N*=3/3 rats) and the hippocampus (*N*=2/3 rats; *Figure 6A*). As a within-subject control, we also stimulated the VMT with a red laser (638 nm). Stimulation with red and blue lasers were randomly interleaved within a recording session and various parameters were explored to identify candidate parameters that would facilitate mPFC-hippocampal coherence.

Optogenetic stimulation of the VMT produced a large negative deflection in the mPFC voltage (*Figure 6D*), but reliably increased mPFC oscillation power that closely matched the VMT stimulation frequency across all animals and sessions (*Figure 6—figure supplement 1*). VMT theta rhythm stimulation increased the power of mPFC theta oscillations across all recording channels from a 64ch silicone probe targeting mPFC lamina (*Figure 6B and C*; see *Figure 6—figure supplement 2* as a

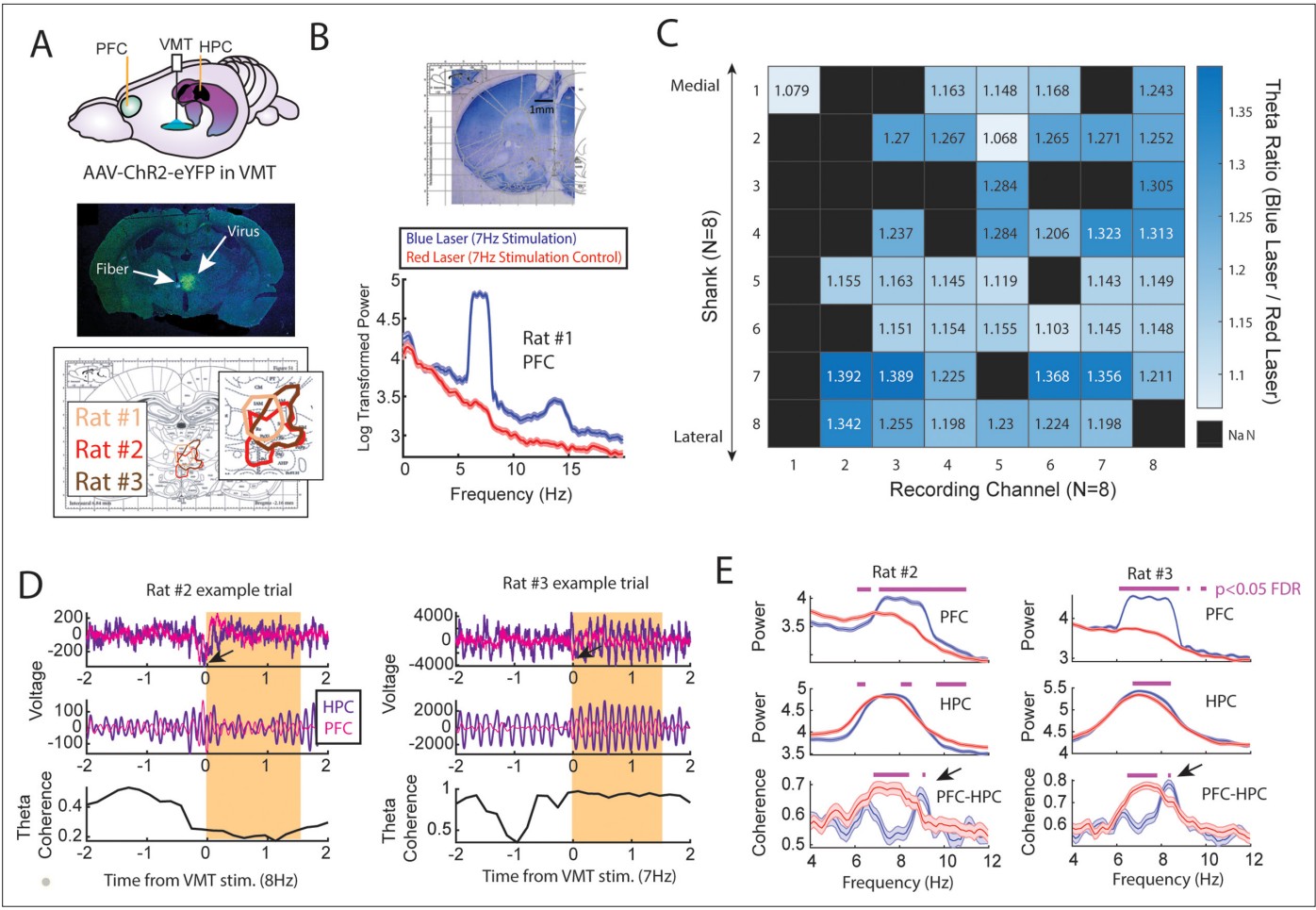

**Figure 6.** Optogenetic activation of the ventral midline thalamus (VMT) increased prefrontal and hippocampal theta power while dynamically adjusting the medial prefrontal cortex (mPFC)-hippocampal theta coherence distribution. (**A**) *Top panel*, Schematic demonstrating recordings from the mPFC and hippocampus with optogenetic activation of the VMT. *Middle panel*, example histological confirmation of fiber implant and viral expression targeting the VMT. *Bottom panel*, Viral expression at similar viral injection coordinates. Notice that all rats showed overlap in viral expression in the nucleus reuniens (brain section overlay from ***Paxinos and Watson, 2006***). (**B**) *Top panel*, Histological confirmation of 64ch silicon probe recordings in the dorsal medial prefrontal cortex. *Bottom panel*, Optogenetic activation of the VMT at 7 Hz produced prefrontal theta rhythms (*N*=83 blue, 88 red laser events; rat #1). (**C**) Ratio of log-transformed mPFC theta (6–9 Hz) power between blue and red laser events across silicon probe shanks and channels. Values >1 indicate that theta during blue laser epochs was stronger than during red laser epochs. 'NaN' represents an excluded channel. See ***Figure 6—figure supplement 2*** for companion figure. Columns represent recording channels per shank, while rows represent shank number from the corresponding medial-lateral placement in the mPFC (**B**). (**D**) Data from rat #2 (*N*=108 blue, 104 red laser events) and rat #3 (*N*=113 blue, 101 red laser events). *Top panel* shows raw local field potential (LFP) traces, *middle panel* shows theta filtered traces (6–9 Hz), while the *bottom panel* shows theta coherence as a function of time. Yellow box shows the stimulation event. Arrows point to observed negative deflects in the LFP signals surrounding VMT stimulation onset. (**E**) Power and coherence analyses performed on data during VMT stimulation (0–1.5 s from laser onset) as a function of frequency (*x*-axis), brain region (row), and rat (*left/right panels*). Both mPFC and hippocampal theta power were increased during VMT stimulation. Coherence between mPFC and hippocampal theta rhythms were reduced or enhanced in a frequency-dependent manner during VMT stimulation. Magenta lines denote p<0.05 following Benjamini-Hochberg p-value corrections for two-sample t-tests between the 6 and 11 Hz range. Data are represented as the mean ± s.e.m.

The online version of this article includes the following figure supplement(s) for figure 6:

**Figure supplement 1.** Sessions recorded with ventral midline thalamus (VMT) stimulation.

**Figure supplement 2.** Prefrontal power spectra across recording shanks and channels during ventral midline thalamus stimulation.

companion figure to ***Figure 6C***). Stimulation of the VMT at 7, 15, or 30 Hz produced clear changes to the mPFC power spectrum, while 4 Hz stimulation was more variable across shanks (***Figure 6—figure supplement 1H***). VMT theta stimulation did not always increase or change hippocampal theta rhythm power, but it often increased or changed the shape of the power spectrum (***Figure 6—figure supplement 1A–G***). Surprisingly, optogenetic activation of the VMT at 7–8 Hz was largely disruptive

to mPFC-hippocampal theta coherence (*Figure 6—figure supplement 1A–G*), but was nonetheless capable of increasing mPFC-hippocampal theta coherence at unexpected frequencies (*Figure 6E*). Specifically, VMT stimulation was better capable of increasing mPFC-hippocampal theta coherence when timed with real-time monitoring of hippocampal oscillations and with sufficient activation. For example, in rat #2, we detected hippocampal oscillation power between 1 and 50 Hz and timed VMT stimulation when 8 Hz power was the strongest frequency. This approach increased mPFC-hippocampal theta coherence in the 9 Hz band (*Figure 6E* and *Figure 6—figure supplement 1G*). Yet, 7 Hz stimulation without timing it to hippocampal oscillatory activity had no effect on mPFC-hippocampal theta coherence (*Figure 6—figure supplement 1F*). Likewise, in rat #3, 7.5 Hz stimulation was sufficient to enhance mPFC-hippocampal theta coherence at 8.3–8.4 Hz at 4.5 mW, but the same was not true at 1 mW power (*Figure 6E* and *Figure 6—figure supplement 1A–C*).

While we expected VMT stimulation to strengthen mPFC-hippocampal theta coherence, these results indicate that square wave optogenetic stimulation of the VMT does not pose a viable approach to strengthen mPFC-hippocampal coherence without consideration of ongoing oscillatory dynamics. Instead, VMT stimulation most effectively produces closely matched oscillations in the mPFC, a finding with interesting implications for diseases characterized by a disrupted thalamic complex (*Elvsåshagen et al., 2021*). Future research should perform a systemic characterization of the parameter space and opsins that allow VMT activation with optogenetics to produce mPFC-hippocampal theta coherence. This study is warranted given the growing hypothesis that the VMT regulates mPFC-hippocampal oscillatory dynamics (*Dolleman-van der Weel et al., 2019*).

## Discussion

Previous research showed that mPFC-hippocampal theta coherence was stronger when memory was used to guide choices (*Jones and Wilson, 2005*; *Benchenane et al., 2010*; *Sigurdsson et al., 2010*; *O'Neill et al., 2013*; *Hallock et al., 2016*), but this conclusion required correlating choice outcome with mPFC-hippocampal theta synchrony. Unlike past work, we manipulated the timing of trial onset relative to the strength of mPFC-hippocampal theta synchrony and as such, the detection of coherence state always preceded choice outcome. Our brain-machine interfacing experiments allowed us to implement various within-subject controls and we showed that trials initiated during states of strong mPFC-hippocampal theta coherence led to better task performance on two separate paradigms.

While we expected this form of long-range theta synchronization to be particularly useful when spatial working memory was used to guide decision-making, we also observed that mPFC-hippocampal theta coherence enhanced the performance of a task that did not require the mPFC, VMT, nor hippocampus for successful performance (*Hallock et al., 2013a*; *Hallock et al., 2013b*; *Shaw et al., 2013*). These findings raise an interesting discrepancy – mPFC-hippocampal theta coherence led to improved task performance on the conditional discrimination task, yet pharmacological inactivation of these structures did not impair task performance. Given that the conditional discrimination task is dependent on the dorsal striatum, it is possible that pharmacological techniques, which work on the scale of minutes, provided time for the brain to adapt to a disrupted mPFC-hippocampal network. In support of this view, *Goshen et al., 2011*, showed that optogenetic suppression of the CA1 on a time scale similar to pharmacological agents, like muscimol, did not impair the retrieval of a contextual fear memory. However, when optogenetic inactivation was temporally specific to the testing phase of the contextual fear memory paradigm, memory retrieval was disrupted. These findings show that the timescale of inactivation impacts the results and conclusions drawn from research, raising the possibility that the mPFC-hippocampal network can indeed be beneficial to the performance of working memory-independent tasks. Future research should be dedicated to testing the causal link of mPFC-hippocampal theta synchronization to choice outcome by implementing procedures similar to what is described here with optogenetic perturbations.

To then characterize the neural dynamics co-occurring with strong mPFC-hippocampal theta coherence events, we tested whether mPFC-thalamic and hippocampal-thalamic interactions changed with strong and weak mPFC-hippocampal theta synchronization events. For these analyses, we focused on the VMT, a structure that is bidirectionally connected with the mPFC and hippocampus and supports mPFC-hippocampal neuronal interactions (*Vertes, 2002*; *McKenna and Vertes, 2004*; *Vertes et al., 2006*; *Hoover and Vertes, 2007*; *Hoover and Vertes, 2012*; *Ito et al., 2015*; *Hallock et al., 2016*). Consistent with mPFC-hippocampal theta coherence reflecting heightened neural coordination across

the brain, VMT theta rhythms showed stronger coherence to mPFC and hippocampal theta rhythms when the mPFC and hippocampus were strongly coherent. Likewise, optogenetic activation of the VMT modulated mPFC and hippocampal theta rhythms, while dynamically altering the way in which these structures were coherent at theta. It should be noted that because hippocampal theta rhythms were already prominent, the effect of VMT stimulation could have appeared less dramatic for hippocampal theta relative to mPFC theta. Nonetheless, our physiological and optogenetic work point toward cortico-thalamic dialogue as a central component of mPFC-hippocampal theta synchronization. Importantly, this latter assertion is supported by anatomy, as the mPFC receives no direct projections from the dHPC (*Jay and Witter, 1991*; *Hoover and Vertes, 2007*), but influences hippocampal neuronal activity via the thalamus (*Ito et al., 2015*). We suspect that the VMT may coordinate mPFC-hippocampal neural interactions through cortico-thalamo-cortical looping mechanisms, as the VMT projects directly to entorhinal cortex neurons that target the CA1 (*Wouterlood, 1991*) and modulates CA1 neurons with concurrent cortical activation (*Dolleman-van der Weel et al., 2017*). Consistent with this hypothesis, mediodorsal thalamus is known to sustain mPFC neuronal activity (*Bolkan et al., 2017*; *Schmitt et al., 2017*), and the VMT supports mPFC firing and mPFC-hippocampal synchronization (*Hallock et al., 2016*; *Jayachandran et al., 2023*).

If mPFC-hippocampal oscillatory synchronization structured cortico-thalamic and cortico-hippocampal neuronal communication, then we would have expected strong theta synchronization events to correlate with mPFC spike entrainment to hippocampal and VMT theta rhythms. When examining all mPFC neurons, we found no differences to spike-LFP synchronization between strong and weak mPFC-hippocampal theta coherence events. Instead, we found a rather small increase to the percentage of theta modulated units in the mPFC. This observation is consistent with recent experimental, modeling, and theoretical work, implicating coherence as a product of communication, rather than a scaffold (*Schneider et al., 2021*; *Vinck et al., 2023*). For example, *Schneider et al., 2021*, showed that LFP signal coherence between a sending and receiving structure can be explained by a sending structures signal power and strength of projectors and can emerge without changes to spike entrainment in the receiving structure. Given that the ventral hippocampus and VMT are necessary for mPFC-hippocampal theta coherence (*O'Neill et al., 2013*; *Hallock et al., 2016*), we suspect that afferents from these structures contribute significantly to mPFC-hippocampal oscillatory synchronization. When taken together, mPFC-hippocampal theta coherence events may represent short temporal periods of neural communication, rather than scaffolding communication. As such, the existing literature combined with our findings strengthen a claim for using patterns of oscillatory synchronization in a therapeutic setting.

Consistent with our work, a recent study found that inducing states of theta synchrony between frontal and temporal regions via transcranial alternating-current stimulation rescued age-related memory impairments in human participants (*Reinhart and Nguyen, 2019*). Our findings suggest that tapping into pre-existing neural dynamics holds significant promise for improving memory. We hypothesize that non-invasive stimulation techniques prior to therapy, paired with synchrony-dependent attention or working memory practice via brain-machine interfacing, could pose a viable intervention to improve cognitive deficits. In closing, the use of brain-machine interfacing holds significant promise for clinical and neuroscientific advance.

## Methods

### Subjects

Subjects were 20 adult (>postnatal day 90) Long Evans Hooded rats. For experiment #1 (*Figures 1–3*), there were 4 adult male and 4 adult female rats with simultaneous mPFC and hippocampus LFP recordings. This sample size was determined a priori using GPower. There was an initial pilot group of rats that were used to troubleshoot the automatic maze and brain-machine interface. These animals were not included in the set of experiments described. For the conditional discrimination experiment (*Figure 4*), three adult rats (two female, one male) were implanted with wires targeting the mPFC and hippocampus. In the analyses from *Figure 5* and *Figure 3—figure supplement 1*, there were six adult male rats. Three adult male rats were selected from *Stout and Griffin, 2020*, due to simultaneous mPFC-VMT-hippocampal recordings (six sessions were removed from one rat due to LFP artifacts). Three adult male rats were selected from *Hallock et al., 2016*, for their mPFC single units,

mPFC LFPs, and hippocampal LFPs (only DA sessions included). For the optogenetic experiment (*Figure 6* and *Figure 6—figure supplement 1*), three male rats received optogenetic virus injections and fiber placement targeting the VMT (two with simultaneous mPFC/hippocampus recordings and one with silicon probe recording from the mPFC). Each rat was placed on mild food restriction (three to four pellets for females, four to five pellets for males) to maintain ~85–90% ad libitum body weight. Rats maintained a 12 hr light/dark cycle in a humidity-controlled colony room. Experimentation was performed during the light cycle (8 am to 5 pm) at approximately the same time each day ±~1 hr. All procedures and protocols were approved by the University of Delaware Institutional Animal Care and Use Committee (IACUC) under protocol AUP 1177. ARRIVE guidelines were followed for this research.

## Automated T-maze

The automated maze was in the shape of a figure eight (*Figure 1A* and *Figure 1—figure supplement 1*) and was purchased from MazeEngineers. The total width of the maze was 136.5 cm and the total length was 74.8 cm. Floor width corresponded to ~12.7 cm, while wall height was ~20.3 cm. The delay zone was a rectangular shape, 12.7 cm wide and 32.7 cm long. Doors were pneumatically controlled via a silent air compressor (SilentAire Super Silent 30-TC), reward delivery (45 mg bio-serv chocolate pellets) was controlled through an automated pellet dispenser, and both were under the control of Arduino powered infrared beams (Adafruit) via custom MATLAB programming (*Figure 1—figure supplement 1*). Walls were placed on the exterior of the maze with distinct visual cues on the left and right choice arms. For two rats on DA, interior walls were placed to improve maze running behavior. These walls were kept in place for the conditional discrimination task. In the delay zone, the south facing wall was lowered on the DA task, but was kept in place for the conditional discrimination task. The maze was surrounded by black curtains with visual cues matching the maze and experimentation occurred in a dimly lit room.

## Brain-machine interface

The brain-machine interface relied upon extracting real-time LFPs, performing coherence analysis, and triggering the choice point door to open according to the magnitude of prefrontal-hippocampal theta coherence. Real-time signal extraction was performed using the Neuralynx Netcom v3.1.0 package code (*NlxGetNewCSCData.m*). Since signals were extracted serially, this code was modified in-house (*NlxGetNewCSCData_2signals.m*) and verified by feeding the same recording lead through two separate recorded channels (*Figure 1—figure supplement 2C*). By iteratively extracting signals into MATLAB from the Neuralynx acquisition system at systematically increasing lags (25–300 ms), we found that waiting 250 ms before extracting new signals provided reliable streaming between the brain and MATLAB (*Figure 1—figure supplement 2A–C*). We then tested the impact of dataset sizes on the strength and the shape of the coherence distribution within the 4–12 Hz range, in real time (*mscohere.m, frequency range = 1:0.5:20*). By linearly increasing the amount of data being analyzed, and calculating coherence over 50 separate intervals from an example rat in real time, we noticed that the dataset sizes strongly impacted the shape of the coherence distribution (*Figure 1—figure supplement 2D–F*), although the effect on coherence magnitude was less robust (*Figure 1—figure supplement 2E*). Since the strongest frequency (4–12 Hz) plateaued at ~8 Hz when analyzing dataset sizes of 1.25 s (*Figure 1—figure supplement 2F*), we chose to use 1.25 s dataset sizes with 250 ms steps forward in time (*Figure 1—figure supplement 2G*). In practice, sampling windows were typically ~1.28 s with ~280 ms overlap and yielded stable coherence estimates across epochs (*Figure 1—figure supplement 2G*). 'Theta coherence' was then defined as 6–11 Hz synchrony according to the frequency × coherence plot (*Figure 1B*). This approach led to clear transitions between high and low magnitude theta coherence (*Figure 1—figure supplement 2H*) indicating that we were accurately tracking coherence in real time. Since brain-machine interfacing handles data acquired in real time, multiple procedures were taken to lower the incidence of signal artifact being used in brain-machine interfacing trials. First, real-time LFPs were detrended by subtracting a third-degree polynomial (*detrend.m*). Then, using a mean and standard deviation calculated over a 10 min recording session that occurred prior to brain-machine interfacing experimentation, LFPs were z-score transformed in real time. During brain-machine interfacing experimentation, real-time detrended LFPs were excluded if >1% of the LFPs were saturated with voltages exceeding 4 std from the mean. Since movement-related artifacts often coincided with strong delta (1–4 Hz) power (*Figure 1—figure supplement 2I*),

we also excluded epochs if delta coherence was stronger than theta coherence. When combined, these approaches isolated coherence distributions with clear theta synchrony (6–11 Hz; *Figure 1B*) and high consistency across rats (*Figure 1—figure supplement 2I and J*).

## Behavior and experimentation

Rats were handled for 5 days in the experimentation room with the lights on and placed on mild food restriction prior to habituation to the automated T-maze. Habituation consisted of 'goal-box' training and 'forced-runs' training. For goal-box training, rats were placed near the reward dispensers for 3 min and were required to eat all pellets within 90 s for 6 trials (3 left dispenser/3 right dispenser). One rat was excluded after not passing goal-box training for 7 consecutive days. For forced-runs, rats traversed the maze to receive a reward at the reward dispenser and were required to eat all rewards for at least 1 day. Rats were often run for multiple forced-runs days. In between traversals, rats waited in the delay pedestal. After maze habituation, rats were trained to perform the continuous alternation (CA) task, where choice alternations were reinforced with chocolate pellets. The CA task was performed 5 days/week for 30 min or 40 trials. Rats were required to perform at 80% accuracy for 2 consecutive days before and after surgery. After surgical recovery, rats were re-handled for 5 days, then placed on the CA task until they again reached criterion. The CA task was implemented to ensure that coherence-contingent choice outcomes (see Brain-machine interface) were not confounded by alternation rule acquisition. Rats were then exposed to the spatial working memory DA task, where in between choice alternations, rats waited in the delay zone for a 5–30 s delay period (randomly distributed). Once rats performed the DA task for 2 consecutive days at 70% accuracy, our brain-machine interface testing occurred. DA task training was implemented to rule out any effect of changing environmental demands on the rats (e.g. the introduction of a delay period), as well as to normalize task performance prior to experimentation. During testing, the experimenter was blinded to trial type and trials were excluded if unexpected events occurred before the choice (e.g. loud noises, fear behavior, twisted recording tether) then saved as a MATLAB variable after the session ended. 20% of trials were experimental (10% high coherence/10% low coherence), while 80% of trials were controls (*Figure 2A*). Exclusion criteria was determined prior to data collection. Trial types were presented psuedo-randomly because high and low coherence trials were required to be presented prior to delay matched control trials. Within blocks of 10 trials, 2 were experimental, 2 were delay matched controls, and 6 were random delays. On a given experimental trial, if rats did not breach the coherence threshold, the trial was initiated after 30 s, and the delay matched control trial was replaced with a random delay. After data collection, LFPs were visualized from trials and trials were marked for exclusion if signal artifacts were present.

For the conditional discrimination experiment, pre-training procedures were similar to what is described above. Rats were randomly assigned to wood-left/mesh-right or wood-right/mesh-left contingencies. Forced-runs training (5 days) included the wood/mesh floor inserts. After recovery from surgery, rats began conditional discrimination training, where a floor insert type dictated the turn direction at the choice (e.g. wood floor insert may require a left turn for a reward). Unlike the DA experiment, brain-machine interfacing began on day 1 of conditional discrimination training to ensure adequate data collection (i.e. it was unclear as to how fast rats could acquire this task on the automatic maze). Data were included for analysis once rats reached a criterion of 70% for 2 consecutive days. The conditional discrimination task was initially designed such that a random sequence of trials was generated where no more than three same-turn directions were rewarded, and so that rats could not receive reward from alternation >60% of the time. Later in data collection, this alternation criterion was lowered to 45% to improve conditional discrimination acquisition. Analysis required that rats performed >70%, alternated <70% of the time, and contributed at least three trials to a session. Unlike the DA dataset, which included high and low coherence trials, the conditional discrimination experiment focused on high coherence trials. The distribution of trial types were as such: 40% high coherence, 40% yoked control (identical delay duration as high coherence trials), and 20% random delay trials. Trial types were distributed in blocks of 10 trials so that corresponding yoked control trials would follow closely to high coherence trials. Per each session, 60% of trials were not controlled by the brain. A trial was initiated if rats did not reach high coherence threshold after 20 s, but rats were required to wait in the delay zone for ~3.5–5 s to segment trials. A computer monitor was placed in the room with the experimenter which provided trial-by-trial instructions (i.e. trial 1: wood-left, trial 2:

mesh-right, etc.). This monitor was also used to monitor LFP data in real time, but the experimenter remained blinded to trial type. Trials were marked for exclusion if unexpected events occurred before the choice (see above).

With respect to data used from *Hallock et al., 2016* (*N*=3 rats) and *Stout and Griffin, 2020* (*N*=3 rats), six rats were trained to perform a DA (*Hallock et al., 2016*) or delayed non-match to position task (*Stout and Griffin, 2020*) to 80% criterion for 2 consecutive days. With respect to the DA task, sessions were included if performance was >75% because rats switched between performing the DA task and the conditional discrimination task. Unlike the brain-machine interfacing experiment where delays varied between 5 and 30 s, rats from *Hallock et al., 2016*, had predictable delay durations of 30 s. With respect to the delayed non-match to position task, sessions were included if performance was >80% (*Stout and Griffin, 2020*). This task differs from delayed alteration in that each trial is comprised of a sample phase, where rats are forced to navigate toward the left or right reward zone, followed by a free choice. Rats were rewarded if their choice was an alternation from the sample phase. Sample phase turn directions were pseudo-randomized to ensure there were no more than three same-turn directions in a row. Data were extracted from delay periods, which separated the sample from choice phase and were 20 s in duration. From choice to sample, there was an intertrial interval of 40 s.

## Surgery

Isoflurane (1–4%) anesthetic was used prior to shaving the scalp and placing rats in the stereotaxic instrument (Kopf). Puralube was applied to rats' eyes throughout the surgery. Lidocaine was injected subcutaneously in the scalp, the scalp was sterilized using chlorhexidine solution (0.2% chlorhexidine gluconate), then incised if rats did not exhibit a kick reflex and eye blink reflex. Bleeding was controlled using hydrogen peroxide. Once the skull was level, bregma was identified, and craniotomies were made above the mPFC and dHPC. mPFC craniotomies were made at +3.1 mm anterior and ±1.0 mm lateral to bregma, while dHPC craniotomies were made at –3.7 mm posterior and ±2.2 mm lateral to bregma. Implants were always on the same hemisphere, but hemispheres were decided pseudo-randomly for each rat in a sex matched manner. For the DA brain-machine interfacing experiment, three right hemisphere (two female, one male) and five left hemisphere (two female, three male) implants were successful. Six rats received cannula implants targeting the contralateral VMT and one rat received electrode implants targeting the contralateral striatum for separate experiments that occurred after the data collected in this report. For the conditional discrimination brain-machine interfacing experiment, all three successful implants were in the right hemisphere. One rat received a 64-channel silicon probe implant (Buzsaki 64L, Neuronexus) at 3.7 mm anterior to bregma and 0.7 mm lateral. A small burr hole was made over the cerebellum for reference wire implants at –10 to –12 mm posterior and ±~1.5 mm lateral to bregma. Five to six bone screws (Fine Science Tools) were implanted as support screws, and one to two bone screws were implanted over the cerebellum for grounding. LFP implants were mounted to the skull using Metabond and the remainder of the microdrive was mounted using dental acrylic (Lang Dental). A shield surrounding the electronic interface board was built using a plastic medicine cup or a copper mesh shielding. Copper mesh shielding was grounded to the same screw grounding the electronic interface board. Rats were given a dose of flunixin (Banamine; 2.5 mg/kg) at least 20 min prior to removal from anesthesia and were placed on ~15 mg children's ibuprofen for a 7-day recovery.

For optogenetic infusions (AAV5-hSyn-ChR2-eYFP) and fiber implants, rat #3 and rat #1 received viral injections at 1.8, 2.4, and 3 mm posterior to bregma. Posterior injections of 2.4 and 3 mm were injected at 2.2 mm lateral and 7.1 mm ventral to brain surface at a 15 degree angle. The injection at 1.8 mm posterior to bregma was injected at 2.2 mm lateral to bregma and 6.6 mm ventral to brain surface at a 15 degree angle. Once the microsyringe was placed into the brain, it sat for 10 min, after which, an injection of.1 µL/min was performed for 2.5 min at each location. The fiber was placed at 2.4 mm posterior to bregma, 2.2 mm lateral to bregma, and 6.8 mm ventral to brain surface from the opposite hemisphere. pAAV-hSyn-hChR2(H134R)-EYFP was a gift from Karl Deisseroth (Addgene plasmid # 26973; http://n2t.net/addgene:26973; RRID:Addgene 26973).

Rat #2 received two separate injections at 1.9 mm posterior to bregma and 1.95 mm lateral to bregma. The microsyringe was placed at 7 mm ventral to brain surface, allowed to settle for 10 min, after which a 2.5 min injection took place at.1 µL/min. Once the injection was complete, the microsyringe

was slowly raised dorsally to 6.7 mm ventral to brain surface, and another injection of 2.5 µL occurred. The fiber was then placed at 6.4 mm from brain surface from the opposite hemisphere.

## Perfusion and histology

Rats were sacrificed with a lethal dose of sodium pentobarbital, then perfused with phosphate-buffered saline (PBS) and 4% paraformaldehyde (PFA). After at least 2 days of post-fixing the implant and brain in 4% PFA, brains were extracted, then cryo-protected in 4% PFA and 30% sucrose (sucrose-PFA). After 1–2 weeks, or when brains sunk to the bottom of the vial, brains were sectioned at 30–50 µm. For implant verification, sections were cresyl stained and imaged with a digital microscope (plugable). To verify viral expression in the optogenetic experiment, sections were gently washed in PBS, covered with ProLong Diamond with DAPI (Life Technologies), coverslipped, then imaged with the Leica Stellaris 8 (supported by NIST 70NANB21H085).

## Electrophysiological recordings

LFPs were recorded on a Neuralynx (Digital Lynx) 64-channel recording system. Neuralynx software (Cheetah) was used to sample LFPs at 2 kHz, and filter LFPs between 1 and 600 Hz. mPFC LFP implants consisted of two stainless steel wires, while dHPC implants consisted of four stainless steel wires, each offset dorsoventrally by ~0.25–0.5 mm. Single units were collected using tetrodes and reported in previous publications (*Hallock et al., 2016*; *Stout and Griffin, 2020*). Spikes were sampled at 32 kHz, bandpass filtered between 0.6 and 6 kHz, and thresholded at 50–75 µV. Clusters were cut using SpikeSort3D with KlustaKwik, then manually curated. Putative pyramidal neurons were selected based on spike waveform and interspike intervals (*Ranck, 1973*).

## Granger prediction

All follow-up spectral analyses were performed on data that was inspected for break-through artifacts. Bivariate Granger prediction was used to assess directionality between PFC and HPC LFPs (code from *Hallock et al., 2016*). Granger prediction is calculated using the variance in errors obtained from univariate and bivariate autoregressions on lagged LFPs. As reported by *Cohen, 2014*:

Univariate: $PFC_t = \sum_{n=1}^{k} a_n PFC_{t-n} + e_t$

Bivariate: $PFC_t = \sum_{n=1}^{k} a_n PFC_{t-n} + \sum_{n=1}^{k} b_n HPC_{t-n} + \epsilon_t$

For each model, $t$ reflects the time point for the LFP data, $k$ reflects the model order, $n$ reflects the lag, $e$ represents the variance not explained by a univariate model, while $\epsilon$ reflects the variance not explained by the bivariate model. Granger prediction in the HPC-to-PFC direction is estimated as such:

$$GC_{HPC->PFC} = log\left(\frac{Var\left[e\right]}{Var\left[\epsilon\right]}\right)$$

Spectral estimates are calculated using Geweke's method in both directions (e.g. PFC-to-HPC and HPC-to-PFC). Bayes information criterion (BIC) was used to estimate model order for each signal and was defined as the lag providing the smallest BIC value (up to 20 lags). The median BIC value across all signals was then rounded and applied to each signal for Granger prediction analysis. For multivariate Granger prediction analysis, we used the freely available MVGC toolbox (*Barnett and Seth, 2014*) downloaded from GitHub. The information criterion and VAR model estimation mode was set to lowess regression and BIC was estimated by testing model orders up to 100 lags with an autocovariance lag of 1000. The same BIC value was used for all signals, as described above. Demeaned signals were fit to a VAR model (*tsdata_to_var.m*), the autocovariance sequence was estimated (*var_to_autocov.m*), and the VAR model was checked for potential error, such as violations to stationarity. Finally, the spectral pairwise causality estimations were calculated (*var_to_spwcgc.m*). Granger prediction and model order estimation was performed on signals of identical size (1.25 s) for both high and low coherence epochs. Code is available on the shared GitHub page (*get_mvgc_parameters.m, get_mvgc_modelOrder.m, get_mvgc_freqGranger.m*).

## Spectral power

Power spectral densities were estimated using the chronux toolbox (*Mitra and Bokil, 2007*) *mtspectrumc* using three tapers with a time-bandwidth product of 2 and *pspectrum.m*. To account for the

1/*f* power law, power spectral estimates were log10 transformed. The frequency corresponding to maximum theta power was defined as 'theta frequency' and performed over the 4–12 Hz frequency range.

## Spike-LFP analyses

Analysis of entrainment was performed over the entire task recording to maximize spike counts. High and low mPFC-hippocampal theta coherence thresholds were determined (see above), then high and low coherence epochs were extracted for each session. Two procedures were implemented for the removal of epochs saturated with recording artifacts. First, large voltage fluctuations were detected on a session-by-session basis by concatenating signal epochs, z-score transforming the concatenated signal, then assigning a standard deviation cut-off value for large voltage events for mPFC, VMT, and hippocampal signals separately. These standard deviation cut-offs were referenced back to a voltage value, and epochs were searched for fluctuating voltage estimates exceeding this threshold. If epochs were saturated by >1% of extreme voltage fluctuations, the epoch was removed. Epochs were also removed if the mPFC or VMT voltages exceeded 3500 mV in the positive or negative direction (tended to fluctuate between –2000 and 2000 mV) in order to minimize the confound of movement-related artifacts on spike-phase comparisons. The cleaned high and low mPFC-hippocampal theta coherence events were then concatenated to create LFP strings. To ensure that spikes were not counted twice in entrainment analysis, the concatenated signal was then filtered for uniquely occurring timestamps.

Spike-phase values were estimated by transforming the filtered signal (4–12 Hz via third-degree Butterworth filtering) via Hilbert transform. Spike-phase values were included if theta was twice the magnitude of delta. Only units with >50 spike-phase estimations during both high and low coherence states were included (*Siapas et al., 2005*; *Jones and Wilson, 2005*; *Hyman et al., 2010*; *Hallock et al., 2016*). Rayleigh's test of non-uniformity was performed and a corresponding p-value was assigned to each neuron representing significant entrainment (*circ_rtest.m*). The mean result length vector (MRL) was calculated using 50 spikes, over 1000 random sampled spike distributions, then taking the average MRL over the 1000 random samples.

Spike-field coherence analysis was used to measure spike-LFP coherence as a function of frequency. Across linearly spaced frequencies (1:20 Hz at 0.5 Hz resolution), complex Morlet wavelets (six cycles) were convolved against the LFP signals. Spike-LFP phase angles were estimated using the analytic signal and calculating the length of the average vector using Euler's formula, defined as SFC (*Cohen, 2017*):

$$SFC_f = \left| \frac{\sum_{k=1}^{N} e^{\sqrt{-1} * \theta_k}}{N} \right|$$

SFC was calculated over each frequency *f*, where *θ* reflects the LFP phase angle per neuron spike timestamp *k* through *N*.

## Behavioral quantification and recording

Behavior was recorded from the rat using two approaches: (1) using a mounted camera sampled at ~30 pixels/s (Cheetah; Neuralynx) that detects LEDs on the recording headstage and (2) by sending TTL pulses to Cheetah when infrared beams were broken on the maze via MATLAB. Time spent to choice was estimated using TTL pulses from the central door opening and from choice point exit (as defined by the infrared beam controlling the closing of the choice point door behind the rat). Behavioral complexity was calculated using the integrated change in absolute angular velocity (IdPhi; code provided by D Redish; *Papale et al., 2012*; *Redish, 2016*) using position data obtained from central door opening to choice point exit. Position data was smoothed using a Gaussian weighted filter (*smoothdata.m*), then velocity in the *x* (*dX*) and *y* (*dY*) dimensions are obtained using a discrete time-adaptive windowing approach (*Janabi-Sharifi et al., 2000*). Phi is defined as the arctangent of *dX* and *dY*, and dPhi is calculated by applying the time-adaptive windowing methodology on the unwrapped Phi estimates. IdPhi is then defined as the integral over the |dPhi| scores. Thus, for each trial, there is one IdPhi score that represents the overall head-movement complexity of the rat. Distance traveled in delay was used to assess whether general mobility differed between experimental and control groups. Position data was extracted from the 1.25 s interval before the choice point door opened (e.g. delay

exit), and total distance traveled was defined as the summation across instantaneous distance, calculated according to the distance equation:

$$Distance\ Traveled\ = \sum_{i=1}^{k} \sqrt{\left(x_{i+1} - x_i\right)^2 + \left(y_{i+1} - y_i\right)^2}$$

where $i$ refers to each video tracking data point through point $k$, and $x/y$ refer to Cartesian coordinates obtained through video tracking. Distance traveled was then normalized across each session to be between 0 and 1, then sorted according to trial type.

## Optogenetics

A Doric laser was programmed with the Neuroscience Studio Software to pulse blue (450 nm) or red (638 nm) lights in a square wave pattern. To test if VMT stimulation could enhance theta synchrony, a variety of stimulation parameters were tested. For theta stimulation, 6–8 Hz frequencies were tested under various conditions. Laser power was tested prior to stimulation and red/blue lasers were matched in terms of mW output. Laser powers varied from 1 to 20 mW. Quiescent states were detected by calculating a ratio between theta and delta LFP power in the hippocampus. Theta:delta ratio values <1 was defined as a candidate quiescent state. Coherence thresholds were also used for the stimulation of the VMT. During awake states, stimulation occurred if theta coherence was greater than the high coherence threshold but less than the low coherence threshold. The data shown in *Figure 6* represent single sessions recorded across animals similarly, with 80–100 red and blue laser stimulation events. The data shown in *Figure 6—figure supplement 1* show various parameter states and their effect on coherence when paired with VMT stimulation across recording sessions. A stimulation event typically lasted 1.5–2 s and then the laser was turned off for 2–6 s. Stimulating the VMT of rat #2 revealed mixed results and sometimes visual observations failed to reveal clear theta in the mPFC, despite clear power increases. Rat #2 received a single anterior-posterior injection of AAV5-hSyn-ChR2-eYFP (see Surgery above).

## Statistics

Each figure panel was considered an independent analysis, and when significant p-values were observed (e.g. p<0.05), they were corrected for multiple comparisons using Bonferroni's method (original p-value multiplied by the number of tests performed) or in some cases using the Benjamini-Hochberg method for many comparisons (i.e. >5 comparisons; *Figure 3H*; *Figure 6*; *Figure 6—figure supplement 1*; code: *fdr_bh.m* by David Groppe). If significance was not observed, the raw p-value was reported. Details regarding statistical testing were reported in the figure captions with information regarding p-value adjustment. Normalized difference scores were defined as such:

$$NormDiff\ = \frac{X - Y}{X + Y}$$

where $X$ and $Y$ refer to within-subject datasets. Normalized difference scores were tested for significance via t-test against a 0-null. Statistical testing was performed in MATLAB and RStudio.

## Acknowledgements

We would like to thank A Garcia for initial optimization of the brain-machine interface and for the creation of conceptual figures which we extended upon (*Figure 1—figure supplement 3*), as well as Z Gemzik, H Rosenblum, D Shaw, A Cestone, J Hoopman, E Walzl, J Mace, and S Adiraju for technical assistance. The brain cartoons were created by W Tang. The rat cartoons were created by G Costa. Both images were downloaded from SciDraw.io. We would like to thank Sylvain Le Marchand for capturing images of viral expression for the optogenetics experiments and we would also like to thank M Stanton, W Kenkel, and T Vickery for feedback on the experiments and early versions of the manuscript. This work was made possible by the Office of Laboratory Medicine. We thank the staff at Neuralynx for technical support. Research was funded by the National Institute of Mental Health under R21 MH117687.

# Additional information

### Funding

| Funder | Grant reference number | Author |
| --- | --- | --- |
| National Institute of Mental Health | R21 MH117687 | Amy L Griffin |

The funders had no role in study design, data collection and interpretation, or the decision to submit the work for publication.

### Author contributions

John J Stout, Data curation, Software, Formal analysis, Validation, Investigation, Visualization, Methodology, Writing – original draft, Writing – review and editing; Allison E George, Data curation, Methodology, Writing – review and editing; Suhyeong Kim, Henry L Hallock, Data curation, Writing – review and editing; Amy L Griffin, Conceptualization, Resources, Supervision, Funding acquisition, Writing – original draft, Writing – review and editing

### Author ORCIDs

John J Stout ⬡ https://orcid.org/0000-0002-0659-1468
Amy L Griffin ⬡ https://orcid.org/0000-0002-4524-2156

### Ethics

All procedures were approved by the University of Delaware Institutional Animal Care and Use Committee (IACUC) under protocol AUP 1177.

Reviewer #1 (Public Review): https://doi.org/10.7554/eLife.92033.3.sa1
Reviewer #2 (Public Review): https://doi.org/10.7554/eLife.92033.3.sa2
Reviewer #3 (Public Review): https://doi.org/10.7554/eLife.92033.3.sa3
Author response https://doi.org/10.7554/eLife.92033.3.sa4

# Additional files

### Supplementary files

• Supplementary file 1. Tables and statistics. (a) Statistics from the delayed alternation brain-machine interfacing experiment from *Figure 2*. (b) Statistics from *Figure 3H* showing change in medial prefrontal cortex (mPFC)-hippocampal theta coherence difference scores (high coherence – low coherence trials) as rats navigated toward and away from the choice point infrared beam. (c) Statistics from *Figure 5* power analysis. (d) Statistics from *Figure 5* coherence analysis. (e) Multivariate granger prediction results (*Figure 5*).

• MDAR checklist

### Data availability

Source data are available on figshare.com: https://doi.org/10.6084/m9.figshare.24599616. Code is available on Github: https://github.com/JohnStout/GriffinLabCode (copy archived at *Rosenblum and Stout, 2024*). Some data for initial brain machine interfacing parameter setting were generated in real-time and are not available (see Figure 1—figure supplement 2B–F).

The following dataset was generated:

| Author(s) | Year | Dataset title | Dataset URL | Database and Identifier |
| --- | --- | --- | --- | --- |
| Stout J, George A, Kim S, Hallock H, Griffin A | 2024 | Using synchronized brain rhythms to bias memory guided-decisions | https://doi.org/10.6084/m9.figshare.24599616 | figshare, 10.6084/m9.figshare.24599616 |

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
