## [Editor Report · eLife assessment]

This study enhances our understanding of the relationship between cortico-hippocampal interactions and behavioral performance. Using an inter-areal coherence metric to gate trial initiation in real time, the authors provide **solid** evidence that links high hippocampal-prefrontal theta coherence to correct performance on spatial working memory and cue-guided decision-making tasks. Although reviewers agreed that the results do not demonstrate causality between hippocampal-prefrontal synchrony and behavioral performance, the findings are viewed as **important** given their potential implications for brain-machine interface applications in humans.

---

## [Referee Report · Reviewer #1 (Public Review)]

Summary:

Information transfer between the hippocampus and prefrontal cortex is thought to be critical for spatial working memory, but most of the prior evidence for this hypothesis is correlational. This study attempts to test this causally by linking trial start times to theta-band coherence between these two structures. The authors find that trials initiated during periods of high coherence led to a dramatic improvement in performance. This applied not only to a spatial working memory task, but also to a cue-guided navigation task, suggesting that coherence in these regions may be a signature of a heightened attentional or preparatory state. The authors supplement this behavioral result with electrophysiological recordings and optogenetic manipulations to test whether ventral midline thalamus is likely to mediate hippocampal-prefrontal coherence.

Strengths:

This study demonstrates a striking behavioral effect; by changing the moment at which a trial is initiated, performance on a spatial working memory task improves dramatically, from around 80% correct to over 90% correct. A smaller but nonetheless robust increase in accuracy was also seen in a texture discrimination task. Therefore, prefrontal-hippocampal synchronization in the theta band may not only be important for spatial navigation, but may also be associated with improved performance in a range of tasks. If these results can be replicated using noninvasive EEG, it would open up a powerful avenue for modulating human behavior.

Weaknesses:

Ventral midline thalamic nuclei, such as reuniens, have reciprocal projections to both prefrontal cortex and hippocampus and are therefore well-situated to mediate theta-band interactions between these structures. However, alternative mechanisms cannot be ruled out by the results of this study. For example, theta rhythms are globally coherent across the rodent hippocampus, and ventral hippocampus projects directly to prefrontal cortex. Theta propagation may depend on this pathway, and may only be passively inherited by VMT.

The optogenetic manipulations are intended to show that theta in VMT propagates to PFC and also affects HPC-PFC coherence. However, the "theta" induced by driving thalamic neurons at 7 Hz is extremely artificial. To demonstrate that VMT is causally involved in coordinating activity across HPC and PFC, it would have been better to optogenetically inhibit, rather than excite, these nuclei. If the authors were able to show that the natural occurrence of theta in PFC depends on activity in VMT, that would be much more convincing test of their hypothesis.

---

## [Referee Report · Reviewer #2 (Public Review)]

A number of previous reports have demonstrated that theta synchrony between the hippocampus (HPC) and prefrontal cortex (PFC) is associated with correct performance on spatial working memory tasks. The main goal of the current study is to examine this relationship by initiating trials either randomly (as has typically been done in previous studies) or during periods of high or low PFC-HPC coherence. To this end, they develop a 'brain-machine interface' (BMI) that provides real-time estimates of PFC-HPC theta coherence, which are then used to control trial onset using an automated figure-eight maze. Their main finding is that choice accuracy is significantly higher on trials initiated when theta coherence is high whereas performance on low coherence trials does not differ from randomly initiated control trials. They also observe a similar result using a non-working memory task in the same maze.

Overall the main experiments (Figures 1-4) are well designed and the BMI approach is convincingly validated. There are also appropriate controls and analyses to rule out behavioral confounds and the results are clearly presented. Although the BMI can not establish a causal relationship between PFC-HPC coherence and behavior, it is helpful for examining how extremes in the distribution of brain states are associated with behavioral performance, something that might be more difficult to examine if trials are initiated randomly. As such, the BMI is an interesting approach for studying the neuronal basis of behavior that could be applied in other fields of neuroscience.

In addition to the behavioral results, the authors also examine what neuronal mechanisms might support enhanced PFC-HPC synchrony (Figures 5-6). Here, they examine the potential contribution of the ventromedial thalamus (VMT) but the results are inconclusive. In particular, the results of optogenetic stimulation of the VMT (Figure 6) show that it both increases and decreases PFC-HPC theta synchrony, depending on the exact frequency range examined. These results are also somewhat preliminary as they come from only 2 animals.

---

## [Referee Report · Reviewer #3 (Public Review)]

Stout et al investigate the link between prefrontal-hippocampal (PFC-HPC) theta-band coherence and accurate decisions in spatial decision making tasks. Previous studies show that PFC-HPC theta coherence positively correlates with task learning and correct decisions but the nature of this relation relies on correlations that cannot show whether coherence leads, coincides or is a consequence of decision making. To investigate more precisely this link, the authors devise a novel paradigm. In this paradigm the rat is blocked during a delay period preceding its choice and they control on a trial-by-trial basis the level of PFC-HPC theta coherence prior to the decision by allowing the rat to access the choice point only at a time when coherence reaches above or below a threshold. The design of the paradigm is very well controlled in many ways. First, using the PFC-HPC theta coherence during the delay period to gate when the rat accesses the choice zone clearly separates this coherence from the behavioural decision itself. Moreover, the behaviour of the animal is similar during high and low coherence periods. Finally, control trials are matched trial-by-trial to the time spent waiting by the rat when gated on theta coherence, which is crucial given that working memory performance depends on delay duration. All these features bolster the specificity of the author's main finding which is that PFC-HPC theta coherence prior to choice making is strongly predictive of accuracy in two tasks : one that requires working memory and another in which behaviour is cue-guided. Although this paradigm does not provide direct causal evidence, it convincingly supports the idea that PFC-HPC theta coherence prior to the behavioural decision is related to correct decision making and is not simply temporally coincidental or a consequence of the decision output.

The authors also investigate the mechanisms behind the increase in PFC-HPC coherence during the task and show that it likely involves the recruitment of a small population of PFC neurons, via interactions with the Ventral Midline Thalamus that could mediate prefrontal/hippocampal dialogue.

A key point of interest is the unexpected result showing a link between theta coherence even in the cue-driven version of the task. As the authors point out, muscimol inhibition of neither PFC nor HPC, nor the ventral midline thalamus impacts performance in this task. This raises the question of why coherence between two areas is predictive of choice accuracy when neither area appears to be causally involved. The authors discuss various options and explanations for this discrepancy which clearly adds to the current debate. Moreover their novel paradigm provides new tools to interrogate when inter-area synchrony is associated with information transfer and when this information is then used to drive behavioural decisions.

---

## [Author Response]

The following is the authors’ response to the original reviews.

**Reviewer #1 (Recommendations For The Authors):**
The brain-machine interface used in this study differs from typical BMIs in that it's not intended to give subjects voluntary control over their environment. However, it is possible that rats may become aware of their ability to manipulate trial start times using their neural activity. Is there any evidence that the time required to initiate trials on high-coherence or low-coherence trials decreases with experience?

This is a great question. First, we designed the experiment to avoid this possibility. Rats were experienced on the sequence of the automatic maze both pre and post implantation (totaling to weeks of pre-training and habituation). As such, the majority of the trials ever experienced by the rat were not controlled by their neural activity. During BMI experimentation, only 10% of trials were triggered during high coherence states and 10% for low coherence states, leaving ~80% of trials not controlled by their neural activity. We also implemented a pseudo-randomized trial sequence. When considered together, we specifically designed this experiment to avoid the possibility that rats would actively use their neural activity to control the maze.

Second, we had a similar question when collecting data for this manuscript and so we conducted a pilot experiment. We took 3 rats from experiment #1 (after its completion) and we required them to perform “forced-runs” over the course of 3-4 days, a task where rats navigate to a reward zone and are rewarded with a chocolate pellet. The trajectory on “forced-runs” is predetermined and rats were always rewarded for navigating along the predetermined route. Every trial was initiated by strong mPFC-hippocampal theta coherence. We were curious as to whether time-to-trial-onset would decrease if we repeatedly paired trial onset to strong mPFC-hippocampal theta coherence. 1 out of 3 rats (rat 21-35) showed a significant correlation between time-to-trial onset and trial number, indicating that our threshold for strong mPFC-hippocampal theta coherence was being met more quickly with experience (Author response image 1). When looking over sessions and rats, there was considerable variability in the magnitude of this correlation and sometimes even the direction (Author response image 1). As such, the degree to which rat 21-35 was aware of controlling the environment by reaching strong mPFC-hippocampal theta coherence is unclear, but this question requires future experimentation.

**Author response image 1. sa4fig1:** Strong mPFC-hippocampal theta coherence was used to control trial onset for the entirety of forced-navigation sessions. Time-to-trial onset is a measurement of how long it took for strong coherence to be met. (A) Time-to-trial onset was averaged across sessions for each rat, then plotted as a function of trial number (within-session experience on the forced-runs task). Rat 21-35 showed a significant negative correlation between time-to-trial onset and trial number, indicating that time-to-coherence reduced with experience. The rest of the rats did not display this effect. (B) Correlation between trial-onset and trial number (y-axis); see (A) across sessions (x-axis). A majority of sessions showed a negative correlation between time-to-trial onset and trial number, like what was seen in (A), but the magnitude and sometimes direction of this effect varied considerably even within an animal.

Is there any evidence that rats display better performance on trials with random delays in which HPC-PFC coherence was naturally elevated?

This question is now addressed in Figure 2—figure supplement 1 and discussed in the section titled “strong prefrontal-hippocampal theta coherence leads to correct choices on a spatial working memory task”.

The introduction frames this study as a test of the "communication through coherence" hypothesis. In its strongest form, this hypothesis states that oscillatory synchronization is a pre-requisite for inter-areal communication, i.e. if two areas are not synchronized, they cannot transfer information. Recent experimental evidence shows this relationship is more likely inverted-coherence is a consequence of inter-areal interactions, rather than a cause. See Schneider et al. (DOI: 10.1016/j.neuron.2021.09.037) and Vinck et al. (10.1016/j.neuron.2023.03.015) for a more in-depth explanation of this distinction. The authors should expand their treatment of this hypothesis in light of these findings.

Our introduction and discussions have sections dedicated to these studies now.

Figure 6 - It would be much more intuitive to use the labels "Rat 1", "Rat 2", and "Rat 3"; the "21-4X" identifiers are confusing.

This was corrected in the paper.

Figure 6C - The sub-plots within this figure are rather small and difficult to interpret. The figure would be easier to parse if the data were presented as a heatmap of the ratio of theta power during blue vs. red stim, with each pixel corresponding to one channel.

This suggestion was implemented in the paper. See Fig 6C. Figure 6—figure supplement 2 now shows the power spectra as a function of recording shank and channel. Also see Figure 6—figure supplement 1H

Ext. Figure 2B - What happens during an acquisition failure? Instead of "Amount of LFP data," consider using "Buffer size".

Corrected.

Ext. Figure 2D-E - Instead of "Amount of data," consider using "Window size"

Referred to as buffer size.

Ext. Figure 2E - y-axis should extend down to 4 Hz. Are all of the last four values exactly at 8 Hz?

Yes. Values plateau at 8Hz. These data represent an average over ~50 samples.

Ext. Figure 2F - consider moving this before D/E, since those panels are summaries of panel F

Corrected.

Ext. Figure 4A - ANOVA tells you that accuracy is impacted by delay duration, but not what that impact is. A post-hoc test is required to show that long delays lead to lower accuracy than short ones. Alternatively, one could compute the correlation between delay duration and proportion correctly for each mouse, and look for significant negative values.

We included supplemental analyses in Figure 1—figure supplement 4.

**Reviewer #2 (Recommendations For The Authors):**
The authors should replace terms that suggest a causal relationship between PFC-HPC synchrony and behavior, such as 'leads to', 'biases', and 'enhances' with more neutral terms.

Causal implications were toned down and wherever “leads” or “led” remains, we specifically mean in the context of coherence being detected prior to a choice being made.

The rationale for the analysis described in the paragraph starting on line 324, and how it fits with the preceding results, was not clear to me. The authors also write at the start of this paragraph "Given that mPFC-hippocampal theta coherence fluctuated in a periodical manner (Extended Fig. 5B [now Figure 3—figure supplement 1])", but this figure only shows example data from 2 trials.

The reviewer is correct. While we point towards 3 examples in the manuscript now, we focused this section on the autocorrelation analysis, which did not support our observation as we noticed a rather linear decay in correlation over time. As such, the periodicity observed was almost certainly a consequence of overlapping data in the epochs used to calculate coherence rather than intrinsic periodicity.

Shortly after the start of the results section (line 112), the authors go into a very detailed description of how they validated their BMI without first describing what the BMI actually does. This made this and the subsequent paragraphs difficult to follow. I suggest the authors start with a general description of the BMI (and the general experiment) before going into the details.

Corrected. See first paragraph of “Development of a closed-loop…”.

In Figure 2C, as expected, around the onset of 'high' coherence trials, there is an increase in theta coherence but this appears to be very transient. However, it is unclear what the heatmap represents: is it a single trial, single session, an average across animals, or something else? In Figure 3F, however, the increase appears to be much more sustained.

The sample size was rats for every panel in this figure. This was clarified at the end of Fig. 3.

In Figure 2D, it was not clear to me what units of measurement are used when the averages and error bars are calculated. What is the 'n' here? Animals or sessions? This should be made clear in this figure as well as in other figures.

The sample size is rats. This is now clarified at the end of Fig 2.

Describing the study of Jones and Wilson (2005), the authors write: "While foundational, this study treated the dependent variable (choice accuracy) as independent to test the effect of choice outcome on task performance." (line 83) It was not clear to me what is meant by "dependent" and "independent" here. Explaining this more clearly might clarify how the authors' study goes beyond this and other previous studies.

The reviewer is correct. A discussion on independent/dependent variables in the context of rationale for our experiment was removed.

**Reviewer #3 (Recommendations For The Authors):**
As explained in the public review, my comments mainly concern the interpretation of the experimental paradigm and its link with previous findings. I think modifying these in order to target the specific advance allowed by the paradigm would really improve the match between the experimental and analytical data that is very solid and the author's conclusions.Concerning the paradigm, I recommend that the authors focus more on their novel ability to clearly dissociate the functional role of theta coherence prior to the choice as opposed to induced by the choice. Currently, they explain by contrasting previous studies based on dependent variables whereas their approach uses an independent variable. I was a bit confused by this, particularly because the task variable is not really independent given that it's based on a brain-driven loop. Since theta coherence remains correlated with many other neurophysiological variables, the results cannot go beyond showing that leading up to the decision it correlates with good choice accuracy, without providing evidence that it is theta coherence itself that enhances this accuracy as they suggest in lines 93-94.

The reviewer is correct. A discussion on independent/dependent variables in the context of rationale for our experiment was removed.

Regarding previous results with muscimol inactivation, I recommend that the authors expand their discussion on this point. I think that their correlative data is not sufficient to conclude as they do that despite "these structures being deemed unnecessary" (based on causal muscimol experiments), they "can still contribute rather significantly" since their findings do not show a contribution, merely a correlation. This extra discussion could include possible explanations of the apparent, and thought-provoking discrepancies that they uncover such as: theta coherence may be a correlate of good accuracy without an underlying causal relation, theta coherence may always correlate with good accuracy but only be causally important in some tasks related to spatial working memory or, since muscimol experiments leave the brain time to adapt to the inactivation, redundancy between brain areas may mask their implication in the physiological context in certain tasks (see Goshen et al 2011).

The second paragraph of the discussion is now dedicated to this.

Possible further analysis :In Extended 4A [now Figure 1—figure supplement 4] the authors show that performance drops with delay duration. It would be very interesting to see this graph with the high coherence / low coherence / yoked trials to see if the theta coherence is most important for longer trials for example.

This is a great suggestion. Due to 10% of trials being triggered by high coherence states, our sample size precludes a robust analysis as suggested. Given that we found an enhancement effect on a task with minimal spatial working memory requirements (Fig. 4), it seems that coherence may be a general benefit or consequence of choice processes. Nonetheless, this remains an important question to address in a future study.

Figure 6: The authors explain in the text that although the effect of stimulation of VMT is variable, overall VMT activation increased PFC-HPC coherence. I think in the figure the results are only shown for one rat and session per panel. It would be interesting to add a figure including their whole data set to show the overall effect as well as the variability.

The reviewer is correct and this comment promoted significant addition of detail to the manuscript. We have added an extended figure (Figure 6—figure supplement 1) showing our VMT stimulation recording sessions. We originally did not include these because we were performing a parameter search to understanding if VMT stimulation could increase mPFC-hippocampal theta coherence. The results section was expanded accordingly.

Changes to writing / figures :The paper by Eliav et al, 2018 is cited to illustrate the universality of coupling between hippocampal rhythms and spikes whereas the main finding of this paper is that spikes lock to non-rhythmic LFP in the bat hippocampus. It seems inappropriate to cite this paper in the sentence on line 65.

We agree with the reviewer and this citation was removed.

Line 180 when explaining the protocol, it would help comprehension if the authors clearly stated that "trial initiation" means opening the door to allow the rat to make its choice. I was initially unfamiliar with the paradigm and didn't figure this out immediately.

We added a description to the second paragraph of our first results section.

Lines 324 and following: the analysis shows that there is a slow decay over around 2s of the theta coherence but not that it is periodical (as in regularly occurring in time), this would require the auto-correlation to show another bump at the timescale corresponding to the period of the signal. I recommend the authors use a different terminology.

This comment is now addressed above in our response to Reviewer #2.

Lines 344: I am not sure why the stable theta coherence levels during the fixed delay phase show that the link with task performance is "through mechanisms specific to choice". Could the authors elaborate on this?

We elaborated on this point further at the end of “Trials initiated by strong prefrontal-hippocampal theta coherence are characterized by prominent prefrontal theta rhythms and heightened pre-choice prefrontal-hippocampal synchrony”

Line 85: "independent to test the effect of choice outcome on task performance." I think there is a typo here and "choice outcome" should be "theta coherence".

The sentence was removed in the updated draft.